# Online Performative Gradient Descent for Learning Nash Equilibria in Decision-Dependent Games

**Zihan Zhu**[*]
Duke University

**Ethan X. Fang**[†]
Duke University

**Zhuoran Yang**[‡]
Yale University

## Abstract

We study multi-agent games within the innovative framework of decision-dependent games, which establishes a feedback mechanism that population data reacts to agents' actions and further characterizes the strategic interactions among agents. We focus on finding the Nash equilibrium of decision-dependent games in the bandit feedback setting. However, since agents are strategically coupled, classical gradient-based methods are infeasible without the gradient oracle. To overcome this challenge, we model the strategic interactions by a general parametric model and propose a novel online algorithm, $\underline{O}$nline $\underline{P}$erformative $\underline{G}$radient $\underline{D}$escent (`OPGD`), which leverages the ideas of online stochastic approximation and projected gradient descent to learn the Nash equilibrium in the context of function approximation for the unknown gradient. In particular, under mild assumptions on the function classes defined in the parametric model, we prove that the `OPGD` algorithm finds the Nash equilibrium efficiently for strongly monotone decision-dependent games. Synthetic numerical experiments validate our theory.

## 1 Introduction

The classical theory of learning and prediction fundamentally relies on the assumption that data follows a static distribution. This assumption, however, does not account for many dynamic real-world scenarios where decisions can influence the data involved. Recent literature on performative classification (Hardt et al., 2016; Dong et al., 2018; Miller et al., 2020) and performative prediction (Perdomo et al., 2020) offers a variety of examples where agents are strategic, and data is performative. For instance, in the ride-sharing market, both passengers and drivers engage with multiple platforms using various strategies such as "price shopping". Consequently, these platforms observe performative demands, and the pricing policy becomes strategically coupled.

In this paper, we explore the multi-agent performative prediction problem, specifically, the multi-agent decision-dependent games, as proposed by Narang et al. (2022). We aim to develop algorithms to find Nash equilibria with the first-order oracle. In this scenario, agents can only access their utility functions instead of gradients through the oracle. Finding Nash equilibria in decision-dependent games is a challenging task. Most existing works primarily focus on finding performative stable equilibria within the single-agent setting, an approach that approximates the Nash equilibrium and is relatively straightforward to compute (Mendler-Dünner et al., 2020; Wood et al., 2021; Drusvyatskiy and Xiao, 2022; Brown et al., 2022; Li and Wai, 2022).

There are two major challenges associated with this problem: (i) the distribution shift induced by performative data, and (ii) the lack of first-order information for the performative gradient. To address these two challenges, we propose a novel online gradient-based algorithm, $\underline{O}$nline $\underline{P}$erformative $\underline{G}$radient $\underline{D}$escent (`OPGD`). In particular, our algorithm employs a general parametric framework to

---

[*]Department of Statistical Science, Duke University. `zihan.zhu@duke.edu`

[†]Department of Biostatistics and Bioinformatics, Duke University. `xingyuan.fang@duke.edu`

[‡]Department of Statistics and Data Science, Yale University. `zhuoran.yang@yale.edu`

37th Conference on Neural Information Processing Systems (NeurIPS 2023).

model the decision-dependent distribution, which provides an unbiased estimator for the unknown gradient, and leverages online stochastic approximation methods to estimate the parametric functions.

## 1.1 Major Contributions

Our work provides new fundamental understandings of decision-dependent games. Expanding upon the linear parametric assumption in Narang et al. (2022), we propose a more comprehensive parametric framework that models decision-dependent distributions of the observed data. We also derive sufficient conditions under this parametric framework that guarantee a strongly monotone decision-dependent game, thereby ensuring a unique Nash equilibrium.

From the algorithmic perspective, we propose OPGD, the first online algorithm to find the Nash equilibrium under linear and kernel parametric models. The core problem in decision-dependent games is estimating the performative gradient. We remark that the existing algorithm only handles the linear case and cannot be extended to the non-linear parametric model (Section 3), and OPGD uses an essentially different method to learn the strategic interaction. To elaborate, under the proposed parametric framework, learning the Nash equilibrium in decision-dependent games can be formulated as a bilevel problem, where the lower level is learning the strategic model and the upper level is finding equilibriums. The OPGD algorithm leverages the ideas of online stochastic approximation for the lower problem and projected gradient descent to learn the Nash equilibrium. Moreover, we acknowledge this learning framework bridges online optimization and statistical learning with time-varying models.

We further prove that under mild assumptions, OPGD converges to the Nash equilibrium. For the linear function class, OPGD achieves a convergence rate of $\mathcal{O}(t^{-1})$, matching the optimal rate of SGD in the strongly-convex setting, where $t$ represents the number of iterations. For the kernel function class $\mathcal{H}$ associated with a bounded kernel $K$, we posit that the parametric functions reside within the power space $\mathcal{H}^\beta$ and evaluate the approximation error of OPGD under the $\alpha$-power norm, where $\alpha$ represents the minimal value that ensures the power space $\mathcal{H}^\alpha$ possesses a bounded kernel. We present the first analysis for online stochastic approximation under the power norm (Lemma 4), in contrast to the classical RKHS norm (Tarres and Yao, 2014; Pillaud-Vivien et al., 2018; Lei et al., 2021). The difference between the RKHS $\mathcal{H}$ and the power space $\mathcal{H}^\beta$ makes the standard techniques fail under the power norm, and we use novel proof steps to obtain the estimation error bound. We demonstrate that OPGD leverages the embedding property of the kernel $K$ to accelerate convergence and achieves the rate of $\mathcal{O}(t^{-\frac{\beta-\alpha}{\beta-\alpha+2}})$. Moreover, OPGD can handle the challenging scenario, where parametric functions are outside the RKHS. See Section 4.2 for more details.

## 1.2 Related Work

**Performative prediction.** The multi-agent decision-dependent game in this paper is inspired by the performative prediction framework (Perdomo et al., 2020). This framework builds upon the pioneering works of strategic classification (Hardt et al., 2016; Dong et al., 2018; Miller et al., 2020), and extends the classical statistical theory of risk minimization to incorporate the performativity of data. Perdomo et al. (2020); Mendler-Dünner et al. (2020); Miller et al. (2021) introduce the concepts of performative optimality and stability, demonstrating that repeated retraining and stochastic gradient methods converge to the performatively stable point. Miller et al. (2021), in pursuit of the performatively optimal point, model the decision-dependent distribution using location families and propose a two-stage algorithm. Similarly, Izzo et al. (2021) develop algorithms to estimate the unknown gradient using finite difference methods. More recently, Narang et al. (2022); Piliouras and Yu (2022) expand the performative prediction to the multi-agent setting, deriving algorithms to find the performatively optimal point.

**Learning in continuous games.** Our work aligns closely with optimization in continuous games. Rosen (1965) lays the groundwork, deriving sufficient conditions for a unique Nash equilibrium in convex games. For strongly monotone games, Bravo et al. (2018); Mertikopoulos and Zhou (2019); Lin et al. (2021) achieve the convergence rate and iteration complexity of stochastic and derivative-free gradient methods. For monotone games, the convergence of such methods is established by Tatarenko and Kamgarpour (2019, 2020). Additional with bandit feedback settings, zeroth-order methods (or derivative-free methods) achieve convergence (Bravo et al., 2018; Lin et al., 2021; Drusvyatskiy et al., 2022; Narang et al., 2022), albeit with slow convergence rates (Shamir, 2013; Lin et al., 2021; Narang et al., 2022). Relaxing the convex assumption, Ratliff et al. (2016); Agarwal et al. (2019); Cotter et al. (2019) study non-convex continuous games in various settings.

**Learning with kernels.** Our proposed algorithm closely relies on stochastic approximation, utilizing online kernel regression for the RKHS function class. Prior research investigates the generalization capability of least squares and ridge regression in RKHS De Vito et al. (2005); Caponnetto and De Vito (2007); Smale and Zhou (2007); Rosasco et al. (2010); Mendelson and Neeman (2010). Meanwhile, extensive works study algorithms for kernel regression. For instance, Yao et al. (2007); Dieuleveut and Bach (2016); Pillaud-Vivien et al. (2018); Lin and Rosasco (2017); Lei et al. (2021) propose offline algorithms with optimal convergence rates under the RKHS norm and $L^2$ norm using early stopping and stochastic gradient descent methods, while Ying and Pontil (2008); Tarres and Yao (2014); Dieuleveut and Bach (2016) design online algorithms with optimal convergence rates. The convergence of kernel regression in power norm (or Sobolev norm) is studied in Steinwart et al. (2009); Fischer and Steinwart (2020); Liu and Li (2020); Lu et al. (2022), with offline spectral filter algorithms achieving the statistical optimal rate under the power norm (Pillaud-Vivien et al., 2018; Blanchard and Mücke, 2018; Lin and Cevher, 2020; Lu et al., 2022).

**Notation.** We introduce some useful notation before proceeding. Throughout this paper, we denote the set $1, 2, \cdots, n$ by $[n]$ for any positive integer $n$. For two positive sequences $\{a_n\}_{n\in\mathbb{N}}$ and $\{b_n\}_{n\in\mathbb{N}}$, we write $a_n = \mathcal{O}(b_n)$ or $a_n \lesssim b_n$ if there exists a positive constant $C$ such that $a_n \leq C \cdot b_n$. For any integer $d$, we denote the $d$-dimensional Euclidean space by $\mathbb{R}^d$, with inner produce $\langle\cdot,\cdot\rangle$ and the induced norm $\|\cdot\| = \sqrt{\langle\cdot,\cdot\rangle}$. For a Hilbert space $\mathcal{H}$, let $\|\cdot\|_{\mathcal{H}}$ be the associated Hilbert norm. For a set $\mathcal{X}$ and a probability measure $\rho_{\mathcal{X}}$ on $\mathcal{X}$, let $\mathcal{L}^2_{\rho_{\mathcal{X}}}$ be the $L^2$ space on $\mathcal{X}$ induced by the measure $\rho_{\mathcal{X}}$, equipped with inner product $\langle\cdot,\cdot\rangle_{\rho_{\mathcal{X}}}$ and $L^2$ norm $\|\cdot\|_{\rho_{\mathcal{X}}} = \sqrt{\langle\cdot,\cdot\rangle_{\rho_{\mathcal{X}}}}$. For any matrix $A = (a_{ij})$, the Frobenius norm and the operator norm (or spectral norm) of $A$ are $\|A\|_F = (\sum_{i,j} a_{ij}^2)^{1/2}$ and $\|A\|_{\text{op}} = \sigma_1(A)$, where $\sigma_1(A)$ stands for the largest singular value of $A$. For any square matrix $A = (a_{ij})$, denote its trace by $\text{tr}(A) = \sum_i a_{ii}$. For any $y \in \mathbb{R}^d$, we denote its projection onto a set $\mathcal{X} \subset \mathbb{R}^d$ by $\text{proj}_{\mathcal{X}}(y) = \arg\min_{x\in\mathcal{X}} \|x - y\|$. The set denoted by $N_{\mathcal{X}}(x)$ represents the normal cone to a convex set $\mathcal{X}$ at $x \in \mathcal{X}$, namely, $N_{\mathcal{X}}(x) = \{v \in \mathbb{R}^d : \langle v, y - x\rangle \leq 0, \ \forall y \in \mathcal{X}\}$. For any metric space $\mathcal{Z}$ with metric $d(\cdot,\cdot)$, the symbol $\mathbb{P}(\mathcal{Z})$ will denote the set of Radon probability measures $\mu$ on $\mathcal{Z}$ with a finite first moment $\mathbb{E}_{z\sim\mu}[d(z, z_0)] < \infty$ for some $z_0 \in \mathcal{Z}$.

## 2  Problem Formulation and Preliminaries

We briefly introduce the formulation of $n$-agent decision-dependent games based on Narang et al. (2022). In this setting, each agent $i \in [n]$ takes the action $x_i \in \mathcal{X}_i$ from an action set $\mathcal{X}_i \subset \mathbb{R}^{d_i}$. Define the joint action $x := (x_1, x_2, \cdots, x_n) \in \mathcal{X}$ and the joint action set $\mathcal{X} = \mathcal{X}_1 \times \cdots \times \mathcal{X}_n \subset \mathbb{R}^d$, where $d := \sum_{i=1}^n d_i$. For all $i \in [n]$, we write $x = (x_i, x_{-i})$, where $x_{-i}$ denotes the vector of all coordinates except $x_i$. Let $\mathcal{L}_i : \mathcal{X} \to \mathbb{R}$ be the utility function of agent $i$. In the game, each agent $i$ seeks to solve the problem

$$\min_{x_i\in\mathcal{X}_i} \mathcal{L}_i(x_i, x_{-i}), \quad \text{where} \quad \mathcal{L}_i(x) := \mathop{\mathbb{E}}_{z_i\sim\mathcal{D}_i(x)} \ell_i(x, z_i). \tag{1}$$

Here $z_i \in \mathcal{Z}_i$ represents the data observed by agent $i$, where the sample space $\mathcal{Z}_i$ is assumed to be $\mathcal{Z}_i = \mathbb{R}^p$ with $p \in \mathbb{N}$ throughout this paper. Moreover, $\mathcal{D}_i : \mathcal{X} \to \mathbb{P}(\mathcal{Z}_i)$ is the distribution map, and $\ell_i : \mathbb{R}^d \times \mathcal{Z}_i \to \mathbb{R}$ denotes the loss function. During play, each agent $i$ performs an action $x_i$ and observes performative data $z_i \sim \mathcal{D}_i(x)$, where the performativity is modeled by the decision-dependent distribution $\mathcal{D}_i(x)$. In the round $t$, the agent $i$ only has access to $z_i^1, \cdots, z_i^{t-1}$ as well as $x^1, \cdots, x^{t-1}$ and seeks to solve the ERM version of (1). We assume the access to the first-order oracle, namely, loss functions $\ell_i$ are known to agents but distribution maps $\mathcal{D}_i$ are unknown.

**Definition 1.** *(Nash equilibrium). In the game (1), a joint action $x^* = (x_1^*, x_2^*, \cdots, x_n^*)$ is a Nash equilibrium (Nash Jr, 1996) if all agents play the best response against other agents, namely,*

$$x_i^* = \arg\min_{x_i\in\mathcal{X}_i} \mathcal{L}_i(x_i, x_{-i}^*) = \arg\min_{x_i\in\mathcal{X}_i} \mathop{\mathbb{E}}_{z_i\sim\mathcal{D}_i(x_i, x_{-i}^*)} \ell_i(x_i, x_{-i}^*, z_i), \quad \forall i \in [n]. \tag{2}$$

In general continuous games, Nash equilibria may not exist or there might be multiple Nash equilibria (Fudenberg and Tirole, 1991). The existence and uniqueness of a Nash equilibrium in a continuous game depend on the game's structure and property. In general, finding the unique Nash equilibrium is only possible for convex and strongly monotone games (Debreu, 1952).

**Definition 2.** *(Convex game). Game (1) is a convex game if sets $\mathcal{X}_i$ are non-empty, compact, convex and utility functions $\mathcal{L}_i(x_i, x_{-i})$ are convex in $x_i$ when $x_{-i}$ are fixed.*

Suppose that utility functions $\mathcal{L}_i$ are differentiable, we use $\nabla_i \mathcal{L}_i(x)$ to denote the gradient of $\mathcal{L}_i(x)$ with respect to $x_i$ (the $i$-th individual gradient). We say the game (1) is $C^1$-smooth if the gradient $\nabla_i \mathcal{L}_i(x)$ exists and is continuous for all $i \in [n]$. Using this notation, we define the gradient $H(x)$ comprised of individual gradients

$$H(x) := (\nabla_1 \mathcal{L}_1(x), \cdots, \nabla_n \mathcal{L}_n(x)).$$

**Definition 3.** *(Strongly monotone game). For a constant $\tau \geq 0$, a $C^1$-smooth convex game (1) is called $\tau$-strongly monotone if it satisfies*

$$\langle H(x) - H(x'), x - x' \rangle \geq \tau \|x - x'\|^2, \quad \text{for all } x, x' \in \mathcal{X}.$$

Note that a $\tau$-strongly monotone game ($\tau > 0$) over a compact and convex action set $\mathcal{X}$ admits a unique Nash equilibrium (Rosen, 1965). According to the optimal conditions in convex optimization (Boyd et al., 2004), this Nash equilibrium $x^*$ is characterized by the variational inequality

$$0 \in H(x^*) + N_{\mathcal{X}}(x^*). \tag{3}$$

We briefly talk about the challenges and our idea of designing the algorithm. In decision-dependent games, the classical theory of risk minimization does not work. The primary obstacles to finding the Nash equilibrium in the game (1) include: (i) the distribution shift induced by performative data, and (ii) the lack of first-order information (gradient). To make it clear, standard methods, such as gradient-based algorithms, necessitate the gradient $H(x)$. However, $H(x)$ is unknown since distributions $\mathcal{D}_i$ are unknown, and estimating $H(x)$ is complex due to the dependency between $\mathcal{D}_i(x)$ and $x$. Mathematically, assuming $C^1$-smoothness, the chain rule directly yields the following expression for the gradient

$$\nabla_i \mathcal{L}_i(x) = \mathop{\mathbb{E}}_{z_i \sim \mathcal{D}_i(x)} \nabla_i \ell_i(x_i, x_{-i}, z_i) + \frac{d}{du_i} \mathop{\mathbb{E}}_{z_i \sim \mathcal{D}_i(u_i, x_{-i})} \ell_i(x_i, x_{-i}, z_i) \Big|_{u_i = x_i}, \tag{4}$$

where $\nabla_i \ell_i(x, z_i)$ denotes the gradient of $\ell_i(x, z_i)$ with respect to $x_i$. The main difficulty is estimating the second term in (4) due to the absence of closed-form expressions.

To estimate the unknown gradient $H(x)$, we impose a parametric assumption on the observed data $z_i$ and model the distribution maps $\mathcal{D}_i$ using parametric functions. Note that the linear parametric assumption was first proposed in Narang et al. (2022). In this paper, we extend this assumption to a general framework and show that under the parametric assumption, the gradient $H(x)$ has a closed-form expression, which yields an unbiased estimator for $H(x)$.

**Assumption 1.** *(Parametric assumption). Suppose there exists a function class $\mathscr{F}$ and $p$-dimensional functions $f_i : \mathcal{X} \to \mathbb{R}^p$ over the joint action set $\mathcal{X}$ such that $f_i \in \mathscr{F}^p$ and*

$$z_i \sim \mathcal{D}_i(x) \iff z_i = f_i(x) + \epsilon_i, \quad \forall i \in [n],$$

*where $\epsilon_i \in \mathbb{R}^p$ are zero-mean noise terms with finite variance $\sigma^2$, namely, $\mathbb{E}\epsilon_i = 0$ and $\mathbb{E}\|\epsilon_i\|^2 \leq \sigma^2$.*

Under Assumption 1, assuming that $f_i$ are differentiable and letting $\mathcal{P}_i$ be the distribution of the noise term $\epsilon_i$, we derive the following expression for the utility functions $\mathcal{L}_i(x) = \mathbb{E}_{z_i \sim \mathcal{D}_i(x)} \ell_i(x, z_i) = \mathbb{E}_{\epsilon_i \sim \mathcal{P}_i} \ell_i(x, f_i(x) + \epsilon_i)$. Then the individual gradient would be $\nabla_i \mathcal{L}_i(x) = \nabla_i \mathbb{E}_{z_i \sim \mathcal{D}_i(x)} \ell_i(x, z_i) = \nabla_i [\mathbb{E}_{\epsilon_i \sim \mathcal{P}_i} \ell_i(x, f_i(x) + \epsilon_i)]$. Consequently, the chain rule directly implies the following expression

$$\nabla_i \mathcal{L}_i(x) = \mathop{\mathbb{E}}_{z_i \sim \mathcal{D}_i(x)} \nabla_i \ell_i(x, z_i) + \left( \frac{\partial f_i(x)}{\partial x_i} \right)^\top \mathop{\mathbb{E}}_{z_i \sim \mathcal{D}_i(x)} \nabla_{z_i} \ell_i(x, z_i), \tag{5}$$

where $\nabla_{z_i} \ell_i(x, z_i)$ denotes the gradient of $\ell_i(x, z_i)$ with respect to $z_i$. Given a joint action $x$, each agent $i$ observes data $z_i \sim \mathcal{D}_i(x)$. Equation (5) suggests the following unbiased estimator for $H(x)$:

$$\widehat{H}(x) := \left( \widehat{\nabla}_i \mathcal{L}_i(x) \right)_{i \in [n]} = \left( \nabla_i \ell_i(x, z_i) + \left( \frac{\partial f_i(x)}{\partial x_i} \right)^\top \nabla_{z_i} \ell_i(x, z_i) \right)_{i \in [n]}. \tag{6}$$

However, direct computation of $\widehat{H}(x)$ is infeasible because $f_i$ are unknown. To overcome this challenge, we approximate the unknown functions $f_i$ with the function class $\mathscr{F}^p$. In fact, the estimation of $f_i$ can be formed as a non-parametric regression problem, namely,

$$\hat{f}_i = \arg\min_{f \in \mathscr{F}^p} \int_{\mathcal{X} \times \mathcal{Z}_i} \|z_i - f(x)\|^2 d\rho_i, \quad \forall i \in [n], \tag{7}$$

where $\rho_i$ is the joint distribution of $(x, z_i)$ induced by $x \sim \rho_{\mathcal{X}}$ and $z_i \sim \mathcal{D}_i(x)$. Here $\rho_{\mathcal{X}}$ is a user-specified sampling distribution on $\mathcal{X}$.

## 3 The OPGD Algorithm

In this section, we consider $\mathscr{F}$ to be the linear and kernel function classes and derive gradient-based online algorithms to find the Nash equilibrium in the game (1), namely, the Online Performative Gradient Descent (OPGD). In each iteration $t$, assuming that $x^t := (x_1^t, \cdots, x_n^t)$ is the output of the previous iteration, OPGD performs the following update for all $i \in [n]$:

  (i) (Estimation update). Update the estimation of $f_i$ by online stochastic approximation for (7).

 (ii) (Individual gradient update). Compute the estimator (6) and perform projected gradient steps

$$x_i^{t+1} = \text{proj}_{\mathcal{X}_i}(x_i^t - \eta_t \widehat{\nabla}_i \mathcal{L}_i(x^t)), \quad \forall i \in [n].$$

**Linear Function Class.** Let $\mathscr{F}$ be the linear function class, namely, $f_i(x) = A_i x$ for $i \in [n]$, where $A_i \in \mathbb{R}^{p \times d}$ are unknown matrices. Then (7) becomes the least square problem $A_i = \arg\min_{A \in \mathbb{R}^{p \times d}} \mathbb{E}_{(u_i,y_i) \sim \rho_i} \|y_i - A_i u_i\|^2$ with random variables $u_i \sim \rho_\mathcal{X}, y_i \sim \mathcal{D}_i(u_i)$. We use the gradient of the least square objective $\|y_i - A_i u_i\|^2$ to derive the online least square update: $A^{\text{new}} \leftarrow A - \nu(Au_i - y_i)u_i^\top$ (Dieuleveut et al., 2017; Narang et al., 2022). In each iteration $t$, we suppose that $A_i^{t-1}$ is the estimation of $A_i$ from the previous iteration, OPGD samples $u_i^t \sim \rho_\mathcal{X}$ and $y_i^t \sim \mathcal{D}_i(u_i^t)$ and performs the following estimation update:

$$\text{(i)} \quad A_i^t = A_i^{t-1} - \nu_t \left( A_i^{t-1} u_i^t - y_i^t \right) (u_i^t)^\top. \tag{8}$$

Recalling (5), the individual gradient is $\nabla_i \mathcal{L}_i(x) = \mathbb{E}_{z_i \sim \mathcal{D}_i(x)} \left[ \nabla_i \ell_i(x, z_i) + A_{ii}^\top \nabla_{z_i} \ell_i(x, z_i) \right]$, where $A_{ii} = \partial f_i(x)/\partial x_i \in \mathbb{R}^{p \times d_i}$ denotes the submatrix of $A_i$ whose columns are indexed by the agent $i$. After step (i), OPGD draws a sample $z_i^t \sim \mathcal{D}_i(x^t)$ and compute the estimator (6) to perform the projected gradient step:

$$\text{(ii)} \quad x_i^{t+1} = \text{proj}_{\mathcal{X}_i} \left( x_i^t - \eta_t \left( \nabla_i \ell_i(x^t, z_i^t) + (A_{ii}^t)^\top \nabla_{z_i} \ell_i(x^t, z_i^t) \right) \right). \tag{9}$$

**Kernel Function Class.** Now we consider $\mathscr{F}$ as the kernel function class, namely, we suppose $f_i \in (\mathcal{H})^p$, where $\mathcal{H}$ is an RKHS induced by a Mercer kernel $K : \mathcal{X} \times \mathcal{X} \to \mathbb{R}$ and a user-specified probability measure $\rho_\mathcal{X}$. By the reproducing property of $\mathcal{H}$, $f_i$ can be represented as $f_i(x) = \langle f_i, \phi_x \rangle_\mathcal{H}$, where $\phi : \mathcal{X} \to \mathcal{H}$ is the feature map, i.e. $\phi_x := K(\cdot, x) \in \mathcal{H}$ for any $x \in \mathcal{X}$. Therefore, (7) becomes the kernel regression $\arg\min_{f \in \mathscr{F}^p} \mathbb{E}_{(u_i,y_i) \sim \rho_i} \|y_i - \langle f, \phi_{u_i} \rangle_\mathcal{H}\|^2$. However, as $\mathcal{H}$ is generally an infinite-dimensional space, the aforementioned regression problem might lead to ill-posed solutions. Consequently, we consider the regularized kernel ridge regression $\arg\min_{f \in \mathscr{F}^p} \mathbb{E}_{(u_i,y_i) \sim \rho_i} \|y_i - \langle f, \phi_{u_i} \rangle_\mathcal{H}\|^2/2 + \lambda_t \|f\|_\mathcal{H}^2$. In each iteration $t$, we suppose that $f_i^{t-1}$ is the estimation of $f_i$ from the previous iteration, the OPGD algorithm samples $u_i^t \sim \rho_\mathcal{X}, y_i^t \sim \mathcal{D}_i(u_i^t)$ and takes gradient steps on the kernel ridge objective $\|y_i^t - \langle f, \phi_{u_i^t} \rangle_\mathcal{H}\|^2/2 + \lambda_t \|f\|_\mathcal{H}^2$, i.e. it takes the online kernel ridge update (Tarres and Yao, 2014; Dieuleveut and Bach, 2016):

$$\text{(i)} \quad f_i^t = f_i^{t-1} - \nu_t \left[ \left( f_i^{t-1}(u_i^t) - y_i^t \right) \phi_{u_i^t} + \lambda_t f_i^{t-1} \right]. \tag{10}$$

We suppose that the kernel $K$ is 2-differentiable, i.e., $K \in C^2(\mathcal{X}, \mathcal{X})$. Define $\partial_i \phi : \mathcal{X} \to \mathcal{H}$ as the partial derivative of the feature map $\phi$ with respect to $x_i$, namely, $\partial_i \phi_x = \partial_i K(x, \cdot) = \partial K(x, \cdot)/\partial x_i$. Steinwart and Christmann (2008, Lemma 4.34) shows that $\partial_i \phi_x$ exists, continuous and $\partial_i \phi_x \in \mathcal{H}$. By the reproducing property $\partial f_i(x)/\partial x_i = \partial \langle f_i, \phi_x \rangle_\mathcal{H}/\partial x_i = \langle f_i, \partial_i \phi_x \rangle_\mathcal{H}$, the individual gradient $\nabla_i \mathcal{L}_i(x)$ has the form $\nabla_i \mathcal{L}_i(x) = \mathbb{E}_{z_i \sim \mathcal{D}_i(x)}[\nabla_i \ell_i(x, z_i) + (\langle f_i, \partial_i \phi_x \rangle_\mathcal{H})^\top \nabla_{z_i} \ell_i(x, z_i)]$. After step (i), OPGD draws a sample $z_i^t \sim \mathcal{D}_i(x^t)$ and performs the projected gradient step:

$$\text{(ii)} \quad x_i^{t+1} \leftarrow \text{proj}_{\mathcal{X}_i} \left( x_i^t - \eta_t \left( \nabla_i \ell_i(x^t, z_i^t) + (\langle f_i^t, \partial_i \phi_{x^t} \rangle_\mathcal{H})^\top \nabla_{z_i} \ell_i(x^t, z_i^t) \right) \right). \tag{11}$$

We remark that the gradient steps $\eta_t, \nu_t$ and regularization terms $\lambda_t$ should be chosen carefully to ensure convergence (see Theorem 2). Specifically, the regularization terms $\lambda_t$ must shift to $0$ gradually. If $\lambda_t$ is a constant, $f_i^t$ in (10) converges to the solution of a regularized kernel ridge regression, which is a biased estimator of $f_i$. Thus (11) fails to converge because the gradient estimation has a constant bias. We present the pseudocode of OPGD for the linear setting as Algorithm 1 and for the RKHS setting as Algorithm 2 in Appendix A.

**Comparison with Narang et al. (2022).** We clarify the difference between OPGD and the Adaptive Gradient Method (AGM) proposed in Narang et al. (2022). To elaborate, AGM samples $z_i^t \sim \mathcal{D}_i(x^t)$ at current the action and let agents play again with an injected noise $u^t$ to obtain $q_i^t \sim \mathcal{D}_i(x^t + u^t)$. The algorithm is based on the fact that $\mathbb{E}[q_i^t - z_i^t | u^t, x^t] = A_i u^t$, which is not related to $x^t$. Thus, $A_i$ can be estimated by online least squares. We remark that $\mathbb{E}[q_i^t - z_i^t | u^t, x^t]$ depends on agents' actions in the non-linear (RKHS) cases, because $\mathbb{E}[q_i^t - z_i^t | u^t, x^t] = f_i(x^t + u^t) - f_i(x^t) = \langle f_i, \phi_{x^t + u^t} - \phi_{x^t} \rangle_{\mathcal{H}}$. Thus, the change of action will bring additional error that makes the estimation fail to converge. In contrast, OPGD lets agents play $u_i^t \sim \rho_{\mathcal{X}}$ to explore the action space and learn the strategic behavior of other agents. OPGD estimates the parametric function by solving the ERM version of (7) using online stochastic approximation (8) and (10). This learning framework can be applied to RKHS and potentially beyond that, such as overparameterized neural networks using the technique of neural tangent kernel (Allen-Zhu et al., 2019).

# 4 Theoretical Results

We provide theoretical guarantees for OPGD in both linear and RKHS settings. We first impose some mild assumptions. Similar assumptions are adopted in Mendler-Dünner et al. (2020); Izzo et al. (2021); Narang et al. (2022); Cutler et al. (2022).

**Assumption 2.** *($\tau$-strongly monotone). The game (1) is $\tau$-strongly monotone.*

**Assumption 3.** *(Smoothness). $H(x)$ is $L$-Lipschitz continuous:*

$$H(x_1) - H(x_2) \le L\|x_1 - x_2\|, \quad \forall x_1, x_2 \in \mathcal{X}.$$

**Assumption 4.** *(Lipschitz continuity in z). Define $\mathcal{D} = \mathcal{D}_1 \times \mathcal{D}_2 \times \cdots \times \mathcal{D}_n : \mathcal{X} \to \mathbb{P}(\mathcal{Z})$, where $\mathcal{Z}$ is the sample space $\mathcal{Z} = \mathcal{Z}_1 \times \mathcal{Z}_2 \times \cdots \times \mathcal{Z}_n$. For all $i \in [n], x \in \mathcal{X}$, there exists a constant $\delta > 0$,*

$$\mathop{\mathbb{E}}_{z \sim \mathcal{D}(x)} \sqrt{\sum_{i=1}^{n} \|\nabla_{z_i} \ell_i(x, z_i)\|^2} \le \delta.$$

**Assumption 5.** *(Finite variance). There exists a constant $\zeta > 0$,*

$$\mathop{\mathbb{E}}_{z_i \sim \mathcal{D}_i(x)} \|\nabla_{i, z_i} \ell_i(x, z_i) - \mathop{\mathbb{E}}_{z_i \sim \mathcal{D}_i(x)} \nabla_{i, z_i} \ell_i(x, z_i)\|^2 \le \zeta^2, \quad \forall i \in [n], \forall x \in \mathcal{X},$$

*where $\nabla_{i, z_i} \ell_i$ denotes the gradient of $\ell_i(x, z_i)$ with respect to $x_i$ and $z_i$.*

We remark that Assumption 3 is the standard smoothness assumption for the utility functions $\mathcal{L}_i(x)$ (Boyd et al., 2004; Nesterov et al., 2018). Since $\mathcal{X}$ is a compact set within $\mathbb{R}^d$, Assumption 4 holds if $\ell_i(x, z_i)$ is Lipschitz continuous in $z_i$ and the gradient $\nabla_{z_i} \ell_i(x, z_i)$ is continuous in $x$, and Assumption 5 holds if $\ell_i(x, z_i)$ is Lipschitz in $x$ and $z_i$ (thus $\nabla_{i, z_i} \ell_i(x, z_i)$ has a bounded norm). Assumption 5 implies that the variances of $\nabla_i \ell_i(x, z_i)$ and $\nabla_{z_i} \ell_i(x, z_i)$ are both bounded by $\zeta^2$ for any $x \in \mathcal{X}$ and $z_i \sim \mathcal{D}_i(x)$. We provide sufficient conditions for Assumption 2 in Appendix B.1.

## 4.1 Convergence Rate in the Linear Setting

We introduce two assumptions necessary to derive theoretical guarantees for the linear function class.

**Assumption 6.** *(Linear assumption). Suppose that the parametric assumption holds (Assumption 1) and $f_i(x) = A_i x$ for $i \in [n]$, where $A_i \in \mathbb{R}^{p \times d}$ are unknown matrices.*

**Assumption 7.** *(Sufficiently isotropic). There exists constants $l_1, l_2, R > 0$ such that*

$$l_1 I \preceq \mathbb{E}_{u \sim \rho_{\mathcal{X}}} u u^\top, \quad \mathbb{E}_{u \sim \rho_{\mathcal{X}}} \|u\|^2 \le l_2, \quad \mathbb{E}_{u \sim \rho_{\mathcal{X}}} \left[ \|u\|^2 u u^\top \right] \preceq R \mathbb{E}_{u \sim \rho_{\mathcal{X}}} u u^\top.$$

Assumption 7 has been studied in the literature on online least squares regression (Dieuleveut et al., 2017; Narang et al., 2022). Essentially, this requires the distribution $\rho_{\mathcal{X}}$ to be sufficiently isotropic and non-singular, and it ensures the random variable $u_i^t \sim \rho_{\mathcal{X}}$ in the online estimation update step (8) can explore all the "directions" of $\mathbb{R}^p$. A simple example that satisfies Assumption 7 is the uniform distribution $\rho_{\mathcal{X}} = \mathcal{U}[0, 1]$, in which case $l_1 = l_2 = 1/3, R = 3/5$.

The next theorem provides the convergence rate of OPGD under the linear setting.

**Theorem 1.** *(Convergence in the linear setting). Suppose that Assumptions 2, 3, 4, 5, 6, and 7 hold. Set $\eta_t = 2/(\tau(t + t_0)), \nu_t = 2/(l_1(t + t_0))$, where $t_0$ is a constant that satisfies $t_0 \ge 2l_2 R / l_1^2$. For*

*all iterations $t \geq 1$, the $x^t$ generated by the* `OPGD` *algorithm in Section 3 for linear function class satisfies*

$$\mathbb{E}\|x^t - x^*\|^2 \leq \frac{(4D_1 + 2D_2(t_0+1)\tau)(t_0+2)^2/(t_0+1)^2}{\tau^2(t+t_0)} + \frac{(t_0+1)^2\|x^1 - x^*\|^2}{(t+t_0)^2}, \quad (12)$$

*where $D_1$ and $D_2$ are constants that*

$$D_1 := 4\zeta^2(1+2(M/(t_0+1)+\sup_{i \in [n]}\|A_i\|_F^2)), \quad D_2 := 2\delta^2 M, \quad M := \frac{2t_0^4 \sum_{i=1}^n \|A_i^0 - A_i\|_F^2}{(t_0+1)^3} + \frac{8nl_2\sigma^2(t_0+2)^2}{l_1^2(t_0+1)^2}.$$

We refer the reader to Appendix C.1 for complete proof. Next, we illustrate the parameters involved in Theorem 1: $\tau$ is the strongly monotone parameter of the game (1), $l_1, l_2, R$ are intrinsic parameters describing the isotropy of the distribution $\rho_{\mathcal{X}}$ (Assumption 7), $\sigma^2$ is the variance of the noise term $\epsilon_i$ defined in Assumption 1, $\zeta$ and $\delta$ describe the continuity of $\ell_i$ (Assumption 4, 5), $t_0$ is a sufficiently large value, $A_i^0$ is the initial estimation of $A_i$, $x^1$ is the initial input. Theorem 1 is a combination of Lemma 2 and Lemma 3, where Lemma 2 is the statistical error of the online approximation step (8) and Lemma 3 is the one-step optimization error of the projected gradient step (9). Theorem 1 implies the convergence rate of `OPGD` in the linear setting is $\mathcal{O}(t^{-1})$, which matches the optimal rate of stochastic gradient descent in the strongly-convex setting.

## 4.2 Convergence Rate in the RKHS Setting

Suppose that $K : \mathcal{X} \times \mathcal{X} \to \mathbb{R}$ is a continuous Mercer kernel, by Mercer's theorem, it has the spectral representation $K = \sum_{i=1}^\infty \mu_i e_i \otimes e_i$, where $\{\mu_i\}_{i=1}^\infty$ are eigenvalues, $\{e_i\}_{i=1}^\infty$ are eigenfunctions, and $\otimes$ denotes the tensor product. Moreover, $\{e_i\}_{i=1}^\infty$ is an orthogonal basis of $\mathcal{L}_{\rho_{\mathcal{X}}}^2$ and $\{\mu_i^{1/2}e_i\}_{i=1}^\infty$ is the orthogonal basis of $\mathcal{H}$, which induces the representation $\mathcal{H} = \{\sum_{i=1}^\infty a_i\mu_i^{1/2}e_i : \{a_i\}_{i=1}^\infty \in \ell^2\}$.

**Definition 4.** *(Power space). For a constant $\alpha \geq 0$, the $\alpha$-power space of an RKHS $\mathcal{H}$ is defined by*

$$\mathcal{H}^\alpha = \left\{\sum_{i=1}^\infty a_i\mu_i^{\alpha/2}e_i : \{a_i\}_{i=1}^\infty \in \ell^2\right\},$$

*equipped with the $\alpha$-power norm $\|\cdot\|_\alpha$ and inner product $\langle\cdot,\cdot\rangle_\alpha$, where $\|\sum_{i=1}^\infty a_i\mu_i^{\alpha/2}e_i\|_\alpha := \left(\sum_{i=1}^\infty a_i^2\right)^{1/2}$ and $\langle\sum_{i=1}^\infty a_i\mu_i^{\alpha/2}e_i, \sum_{i=1}^\infty b_i\mu_i^{\alpha/2}e_i\rangle_\alpha = \sum_{i=1}^\infty a_ib_i$.*

We remark that: (i) $\mathcal{H}^1 = \mathcal{H}$ and $\mathcal{H}^\alpha \subset \mathcal{H}^\beta$ for any $\alpha > \beta$, (ii) $\|\cdot\|_1 = \|\cdot\|_{\mathcal{H}}$ and $\|\cdot\|_0 = \|\cdot\|_{\rho_{\mathcal{X}}}$, and (iii) $\mathcal{H}^\alpha$ is an RKHS on $\mathcal{X}$ with kernel $K^\alpha := \sum_{i=1}^\infty \mu_i^\alpha e_i \otimes e_i$ and measure $\rho_{\mathcal{X}}$. We review more properties of RKHS and power spaces in Appendix Sections B.3 and B.4.

We present assumptions on the kernel function class, similar assumptions can be found in the literature on kernel regression and stochastic approximation (Caponnetto and De Vito, 2007; Steinwart et al., 2009; Dicker et al., 2017; Pillaud-Vivien et al., 2018; Fischer and Steinwart, 2020).

**Assumption 8.** *(Source condition). Suppose Assumption 1 holds and there exists an RKHS, $\mathcal{H}$, with a bounded differentiable Mercer kernel, $K$, and constants $\beta, \kappa > 0$ such that $\sup_{x \in \mathcal{X}} K(x,x) \leq \kappa^2$ and $f_i \in (\mathcal{H}^\beta)^p$ for all $i \in [n]$.*

**Assumption 9.** *(Embedding property). There exist constants $\alpha \in (0,1]$, $A > 0$ such that $K^\alpha(x,x) = \sum_{i=1}^\infty \mu_i^\alpha e_i^2(x) \leq A^2$, for all $x \in \mathcal{X}$.*

**Assumption 10.** *(Lipschitz kernel). Suppose Assumption 9 holds and there exists $\xi > 0$ such that $\|\partial_i\phi_x^\alpha\|_\alpha \leq \xi$ for any $i \in [n]$ and $x \in \mathcal{X}$, where $\phi_x^\alpha : \mathcal{X} \to \mathcal{H}^\alpha$ is the feature map of the kernel $K^\alpha$.*

Assumption 8 holds when $K$ is bounded, differentiable, and each coordinate of parametric functions $f_i$ lies in the power space $\mathcal{H}^\beta$. When $\beta < 1$, Assumption 8 includes the challenging scenario, namely, $f_i \notin (\mathcal{H})^p$. Assumption 9 holds if there exists a power space $\mathcal{H}^\alpha$ such that the kernel $K^\alpha$ is bounded. Thus, Assumption 9 holds with $\alpha = 1$ for any bounded kernel $K$. We further propose Proposition 1 as sufficient conditions for the embedding property following Mendelson and Neeman (2010). Recalling the definition of partial derivative $\partial_i\phi^\alpha : \mathcal{X} \to \mathcal{H}^\alpha$ (Section 3), Assumption 10 holds if $\partial_i\partial_{i+d}K^\alpha(x,x) = \|\partial_i\phi_x^\alpha\|_\alpha^2 \leq \xi^2$ for any $x \in \mathcal{X}$, i.e. it holds for any Lipschitz kernel $K^\alpha$.

**Proposition 1.** *(Sufficient conditions for Assumption 9) Suppose there exist constants $C, D, p > 0$ and $q \in (0, 1)$ such that*

$$\sup_{i \in \mathbb{N}} \mu_i^p \|e_i\|_\infty \leq C \quad and \quad \mu_i \leq D i^{-1/q},$$

*where $\|\cdot\|_\infty$ denotes the $L^\infty$ norm. Then Assumption 9 holds for any $\alpha > 2p + q$.*

Proposition 1 follows from the inequality: $\sup_{x \in \mathcal{X}} K^\alpha(x, x) = \sup_{x \in \mathcal{X}} \sum_{i=1}^\infty (\mu_i^p e_i(x))^2 \mu_i^{\alpha - 2p} \leq C^2 D^{\alpha - 2p} \sum_{i=1}^\infty i^{-(\alpha - 2p)/q} < \infty$. We refer readers to Appendix B.4 for examples that satisfy these assumptions. Next, we provide the convergence of the proposed algorithm under the RKHS setting.

**Theorem 2.** *(Convergence in the RKHS setting). Suppose that Assumptions 2, 3, 4, 5 hold, Assumption 8 holds for some $\beta \in (0, 2]$, and Assumptions 9, 10 hold for some $\alpha \in (0, 1]$ and $\alpha < \beta$. For all iterations $t \geq 1$ and positive constant $a$, define $\bar{t} = t + t_0$, where $t_0$ is a constant that satisfies $t_0 \geq (a\kappa^2 + 1)^2$. Set the gradient steps and regularization terms as*

$$\eta_t = (\tau \bar{t})^{-1}, \quad \nu_t = a \cdot \bar{t}^{-\frac{\beta - \alpha + 1}{\beta - \alpha + 2}}, \quad \lambda_t = a^{-1} \cdot \bar{t}^{-\frac{1}{\beta - \alpha + 2}}.$$

*If $a < \sqrt{(\beta - \alpha + 2)/(\beta - \alpha)}(t_0 + 1)/(t_0 + 2)\kappa^{\alpha - 2}A^{-1}$, the $x^t$ generated by the OPGD algorithm in Section 3 using kernel $K$ for online estimation steps (10) and projected gradient steps (11) satisfies*

$$\mathbb{E}\|x^t - x^*\|^2 \lesssim \mathcal{O}(t^{-\frac{\beta - \alpha}{\beta - \alpha + 2}}). \tag{13}$$

See Appendix C.2 for complete proof. We remark that bounding the estimation error $\mathbb{E}\|f_i^t - f_i\|_\gamma^2$ under the $\gamma$-power norm for some $\gamma \in [\alpha, \beta)$ and $\gamma \leq 1$ (Lemma 4) plays a key role in the proof of Theorem 2, and we regard this lemma as the most challenging part of our theory. Lemma 4 extends the classical theory of online stochastic approximation under the RKHS norm $\|\cdot\|_{\mathcal{H}}$ into a continuous scale. For $\beta > 1$ and $\gamma = 1$, this rate would be $\mathcal{O}(t^{-(\beta-1)/(\beta+1)})$ and matches the optimal rate under the RKHS norm Ying and Pontil (2008); Tarres and Yao (2014). If the embedding property (Assumption 9) holds for some $\alpha < 1$, we choose $\gamma = \alpha$ to achieve a faster rate $\mathcal{O}(t^{-(\beta-\alpha)/(\beta-\alpha+2)})$ (which further leads to Theorem 2). To prove Lemma 4, we derive novel proof steps, where we decouple the power norm by considering semi-population iteration and recursive decomposition, we refer the reader to Appendix E.1 for more explanations.

We demonstrate the parameters involved in Theorem 2. Parameters $\alpha, \beta, \kappa, \tau, A$ are intrinsic: $\beta, \kappa, A$ are determined by source condition (Assumption 8), $\alpha$ is determined by embedding property (Assumption 9), and $\tau$ is the strongly monotone parameter. Parameters $a, t_0$ are user-specified: $t_0$ is a sufficiently large value, $a$ is characterized by the inequality $a < \sqrt{(\beta - \alpha + 2)/(\beta - \alpha)}(t_0 + 1)/(t_0 + 2)\kappa^{\alpha-2}A^{-1}$ when $t_0$ is determined, a smaller $a$ leads to a larger constant term in (13). Theorem 2 implies that OPGD leverages the embedding property (Assumption 9) to obtain better convergence rates. For any bounded kernel, Assumption 9 holds for $\alpha = 1$, thus Theorem 2 guarantees the rate $\mathcal{O}(t^{-\frac{\beta-1}{\beta+1}})$. Moreover, suppose that the kernel satisfies some good embedding property, that is, $\alpha < 1$, since larger $\beta - \alpha$ leads to faster convergence rates. In that case, we obtain a better rate $\mathcal{O}(t^{-\frac{\beta-\alpha}{\beta-\alpha+2}})$ by setting the gradient steps and regularization terms $\nu_t, \lambda_t$ corresponding to $\alpha, \beta$. Besides, OPGD can handle the challenging scenario ($f_i \notin (\mathcal{H})^p$ if $\beta < 1$) when the embedding property of kernel holds for $\alpha < \beta$.

## 5 Numerical Experiments

In this section, we conduct experiments on decision-dependent games in both the linear and the RKHS settings to verify our theory. All experiments are conducted with Python on a laptop using 14 threads of a 12th Gen Intel(R) Core(TM) i7-12700H CPU.

**Basic Setup.** We consider two-agent decision-dependent games with 1-dimensional actions. Namely, for all $i \in [2]$, define the game

$$\min_{x \in \mathcal{X}} \mathcal{L}_i(x), \quad \text{where} \quad \mathcal{L}_i(x) := \mathbb{E}_{z_i \sim \mathcal{D}_i(x)} \ell_i(x, z_i), \tag{14}$$

where $\mathcal{X} = [0, 1] \times [0, 1]$, $x \in \mathcal{X}$, $z_i \in \mathbb{R}$, and $\ell_i(x, z_i)$ is the loss function to be determined. Let the distribution map be $\mathcal{D}_i(x) \sim \mathcal{N}(f_i(x), 0.2)$, where $f_i$ is the parametric function determined by the specific function class. Then the game (14) follows the parametric assumption (Assumption 1) with $z_i = f_i(x) + \epsilon_i$, where $\epsilon_i \sim \mathcal{N}(0, 0.2)$ the independent Gaussian noise term.

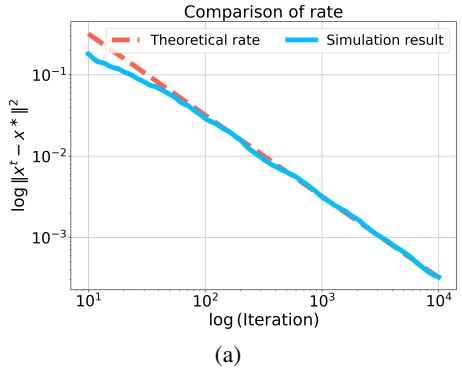
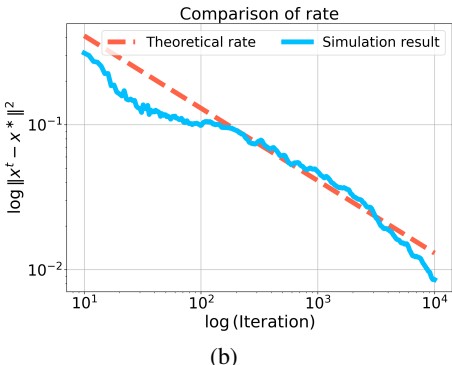

(a)                                          (b)

Figure 1: **(a) Linear setting:** The X-axis represents the iteration from $1$ to $10,000$, while the Y-axis represents the norm-squared error of $x^t$ to the Nash equilibrium $x^* = (1/2, 1)$, averaged over 20 random seeds. The blue solid line represents the output of OPGD and the orange dashed line represents the theoretical rate $\mathcal{O}(t^{-1})$. **(b) RKHS setting:** The X-axis represents the iteration from $1$ to $10,000$, while the Y-axis represents the norm-squared error to the Nash equilibrium $x^* = (1/2, 1/2)$, averaged over 400 random seeds. The blue solid line represents the output of OPGD and the orange dashed line represents the theoretical rate $\mathcal{O}(t^{-1/2})$.

**Linear function class.** Let the loss function be $\ell_i(x, z_i) = -z_i + x_i^2$ and set the linear parametric function as $f_1(x) = x_1$ and $f_2(x) = 2x_2$, namely, the parametric model is $z_i = A_i x + \epsilon_i$ where $A_1 = [1 \; 0]$ and $A_2 = [0 \; 2]$. The the game (14) has the gradient $H(x) = (2x_1 - 1, 2x_2 - 2)$, therefore, the game (14) is convex, $C^1$-smooth, 1-strongly monotone and the Nash equilibrium is $x^* = (1/2, 1)$. We set the sampling distribution as $\rho_{\mathcal{X}} = \mathcal{U}[0, 1] \times \mathcal{U}[0, 1]$, the initial point as $x^0 \sim \rho_{\mathcal{X}}$, and the initial estimation as zero. Moreover, letting $t_0 = 10$, we set the gradient step sizes as $\eta_t = 6/(t + t_0), \nu_t = 6/(t + t_0)$.

**Kernel function class.** Let $\mathcal{X} = [0, 1] \times [0, 1]$, $\rho_{\mathcal{X}} = \mathcal{U}[0, 1] \times \mathcal{U}[0, 1]$, and define the kernel $Q((x_1, x_2), (y_1, y_2)) = K(x_1, y_1) \cdot K(x_2, y_2)$ as the product kernel of $K(x, y) = 40B_4(\{x - y\})$. Suppose that $\mathcal{H}$ is the RKHS on $\mathcal{X}$ induced by the kernel $Q$ and the distribution $\rho_{\mathcal{X}}$. Set the parametric function as the product of two 3-order Bernoulli polynomials, namely, $f(x_1, x_2) = B_3(x_1) \cdot B_3(x_2) = (x_1^3 - 3x_1^2/2 + x_1/2) \cdot (x_2^3 - 3x_2^2/2 + x_2/2)$. Set $\ell_i(x, z_i) = -z_i + \cos(2\pi x_1)\cos(2\pi x_2) - x_i + x_i^2$ and let $f_i(x) = \cos(2\pi x_1)\cos(2\pi x_2)$ for $i \in [2]$. Then the gradient of this game is $H(x) = (2x_1 - 1, 2x_2 - 1)$, thus, this game is convex, $C^1$-smooth, 1-strongly monotone and the Nash equilibrium is $x^* = (0.5, 0.5)$. Following Example 1, Assumption 8, 9, 10 hold for any $\beta > 1$ and any $\alpha > 1/4$. Set $t_0 = 10, a = 7, \eta_t = 6/(t+t_0), \nu_t = a/(t+t_0)^{3/4}$, and $\lambda_t = 1/(a(t+t_0)^{1/4})$. Following Theorem 2, the convergence rate is $\mathcal{O}(t^{-1/2})$.

**Results.** We perform experiments for both parametric settings to verify the convergence rates and compare the theoretical and simulated rates, as shown in Figure 1, where both X and Y axes take the log scale. Figure 1(a) shows the converge rate of the linear setting within $10,000$ iterations, the simulated rate matches our prediction, i.e. it is close to $\mathcal{O}(t^{-1})$. Figure 1(b) shows the convergence rate of the RKHS setting, it implies that the simulated rate is close to the theoretical rate $\mathcal{O}(t^{-1/2})$ when the iteration $t$ is larger than $1,000$. These results validate Theorems 1 and 2.

## 6 Conclusion and Discussion

In this paper, we study the problem of learning Nash equilibria in multi-agent decision-dependent games with access to the first-order oracle. We propose a parametric assumption to handle the distribution shift and develop a novel online algorithm OPGD in both the linear and RKHS settings. We derive sufficient conditions to ensure the decision-dependent game is strongly monotone under the parametric assumption. We show that OPGD converges to the Nash equilibrium at a rate of $\mathcal{O}(t^{-1})$ in the linear setting and $\mathcal{O}(t^{-\frac{\beta - \alpha}{\beta - \alpha + 2}})$ in the RKHS setting.

## Acknowledgement

We would like to thank the area chair, and three anonymous reviewers for their very helpful comments, which help us substantially improve our paper. E. X. Fang is partially supported by NSF DMS-2230795 and DMS-2230797.

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
