# OpenReview forum: "Online Performative Gradient Descent for Learning Nash Equilibria in Decision-Dependent Games"
_NeurIPS.cc/2023/Conference — NeurIPS 2023 poster_

### Official Review · Reviewer_wDhV · 2023-06-25

**Soundness:** 3 good
**Presentation:** 3 good
**Contribution:** 2 fair
**Rating:** 6
**Confidence:** 3

**Summary:**

This paper studies the problem of multi-agent performative game. The authors generalize the existing assumption in (Narang et al., 2022) from linear cases to generalized linear cases, and proposed algorithms consisting online stochastic approximation of the link function and individual gradient updates. They prove convergence rates of linear function class and kernel function class, with numerical experiments validate their theory.

**Strengths:**

The problem statement is clear, and the solution is intuitive. The authors were the first to discuss how to estimate the unknown structure of the distribution in multi-agent performative games, which is a definitely important topic in the machine learning literature.

**Weaknesses:**

The authors should elaborate more about their technical contribution. The current recipe of the proofs mostly follows (Narang et al., 2022), with the accurate parameter of distribution map replaced by an estimated version, which further generates similar error terms that can be controlled via similar techniques in bounding existing terms. It would be better if the authors could emphasize their technical contributions on the way to their theoretical results.

**Questions:**

Some notations are wrong or at least confusing:

  Page 4, assumption 1: Is the second $\mathfrak{F}^{p}$ actually $\mathfrak{F}$?

  Page 4, Line 162: Would it be better to use another notation (i.e., $\hat{f}_i$) on the left-hand-side in (7), since $f_i$ already denotes the ground-truth function of player $i$?


**Limitations:**

No.

---

> ### Author Rebuttal · Authors · 2023-08-09
>
> Thanks for your review and suggestions. We have explained our novelty over Narang et al. (2022) in the global response. We refer the reviewer to **Novel I** for the comparison of OPGD and Algorithm 1 and Narang et al. (2022). We refer to **Novel II** for the explanation that extending the linear algorithm into the non-linear (RKHS) case is not trivial and our approach (introduce the dynamic regularization term $\lambda_t$). We summarize our contribution to the convergence analysis under the power norm in **Novel III** and present a detailed explanation for Lemma 4 and Theorem 2 in **Interpretation of Results in the RKHS Case**. We will address the remaining questions in the following text.
>
> **"The authors should elaborate more about their technical contribution $\cdots$":**
> We apologize that the technical contribution was not highlighted clearly in the initial submission. Although the proof strategy for the linear case is similar to Narang et al. (2022), the proof in the RKHS case (Theorem 2) is non-trivial, specifically, we derive the first convergence analysis for online stochastic approximation under the power norm in Lemma 4 (which is the most challenging part of our work).
>
> We will demonstrate the technical challenges and our approach for deriving the estimation error bound (i.e. Lemma 4, which is the key lemma to obtain Theorem 2) in the following text. To begin with, we refer the reviewer to the explanation of our results (**Novel III**) in the global response for better understanding.
>
> **Technical Challenges of Lemma 4.** With the notations set out in Appendix E, the primary obstacle in establishing power norm bounds for iteration (75) ($f_t$) under the norm $\lVert\cdot\rVert_\gamma$ comes from the difference between the power space $\mathcal{H}^\gamma$ and the RKHS $\mathcal{H}$. To explain further, the standard technique to derive error bounds under $\lVert\cdot\rVert_\mathcal{H}$ decomposes the error $f-f_t$ using the operator $I-\nu_t(L_t+\lambda_tI)$ (as seen in Theorem 3.1 in [1]), with the analysis based on the fact that $I-\nu_t(L_t+\lambda_tI)$ is a contraction map in $\mathcal{H}$. In more detail, $L_t$ is compact, self-adjoint, and positive-semidefinite on $\mathcal{H}$, leading to $\lVert I-\nu_t(L_t+\lambda_tI)\rVert_{\mathcal{H}\rightarrow\mathcal{H}}\leq 1-\nu_k\lambda_k$. However, this operator does not have the same property in the power space $\mathcal{H}^\gamma$. By the definition $L_t=\phi_t^*\phi_t$, for any $h_1,h_2\in\mathcal{H}^\gamma$, we have $L_t(h_i) = h_i(x^t)\phi_t=\langle h_i,\phi_t^\gamma\rangle_\gamma\phi_t$ and $\langle L_th_1,h_2\rangle_\gamma\neq\langle h_1,L_th_2\rangle_\gamma$, and therefore, $L_t$ isn't self-adjoint or positive-definite on $\mathcal{H}^\gamma$. This makes the spectral norm $\lVert I-\nu_t(L_t+\lambda_tI)\rVert_{\mathcal{H}^\gamma\rightarrow\mathcal{H}^\gamma}$ possibly larger than 1.
>
> **Our Approach.** To overcome the aforementioned obstacle, our approach has four steps:
> 1. Given that $L_t$ is stochastic and challenging to analyze under the power norm, we leverage the covariance operator $L_K$ (which is the mean operator of $L_t$) and decompose the error $f-f_t$ by $L_K$ and $L_K-L_t$. We observe that $L_K$ has some good properties under the power norm (see step 3) and $L_K-L_t$ can be further decomposed by semi-stochastic decomposition (step 4). We define the semi-stochastic population iteration (76) ($g_t$), which replaces $L_t$ with $L_K$ and keeps the stochastic term $y_t\phi_t$. We further decompose the error $f-f_t$ by $f-f_{\lambda_t}$, $f_{\lambda_t}-g_t$, and $f_t-g_t$  (Lemma 6 and Remark 2).
> 2. For the term $\lVert f-f_{\lambda_t}\rVert_\gamma$, we obtain the error bound by spectral decomposition (Lemma 7). We remark that $f_{\lambda_t}$ is the solution of the kernel ridge regression with regularization $\lambda_t$ (line 763) and $f$ minimizes the unregularized kernel regression. Intuitively, $\lVert f-f_{\lambda_t}\rVert_\gamma$ converges to zero if $\lambda_f\rightarrow0$.
> 3. To analyze the term $g_t-f_{\lambda_{t}}$, we derive the martingale decomposition (81) and notice that $L_K$ is commutative with the operator $\Pi_i^j$. This commutativity allows us to use Lemma 18, expressed as $\lVert f_{\lambda_t}-g_t\rVert_\gamma = \lVert L_K^{(1-\gamma)/2}(f_{\lambda_t}-g_t)\rVert_{\mathcal{H}}$, to transfer the analysis from power norm bounds to RKHS norm bounds (Lemma 9).
> 4. For the term $f_t-g_t$, which is primarily characterized by the sampling error operator $L_K-L_t$, we use semi-stochastic decomposition (86) recursively to decompose this term. Specifically, we observe that the iteration structure of $f_t-g_t$ is identical to that of $f_t$. We thus consider its semi-stochastic population iteration $r_t^{(0)}$. Further, we examine the error $f_t-g_t-r_t^{(0)}$ and use semi-stochastic decomposition recursively to define a noise process ${r_{t}^{(k)}}$ (86). We then prove that $f_t-g_t$ can be written as a finite sum of ${r_{t}^{(k)}}$ (Lemma 16) and obtain the power norm bound of each $r_{t}^{(k)}$ (Lemma 14).
>
> **"Page 4, assumption 1: Is the second $\cdots$":**
> We apologize for the confusion. The function $f_i$ defined in Assumption 1 is a p-dimensional vector function, here we assume each coordinate of $f_i$ belongs to the function class $\mathscr{F}$. We will revise this sentence for better understanding.
>
> **"Page 4, Line 162: Would it $\cdots$":**
> Thanks for your suggestion! We will use $\hat{f}_i$ instead of $f_i$ in the revision.
>
> [1] Tarres, P. and Yao, Y. Online learning as stochastic approximation of regularization paths: Optimality and almost-sure convergence. IEEE Transactions on Information Theory, 2014.

---

> > ### Comment · Reviewer_wDhV · 2023-08-18
> >
> > Thanks for your detailed response. The rebuttal do make me feel that the novelty of the paper is more than what I've recognized, and I have raised my score.
> >
> > In particular, I agree with the saying in Novel I in global rebuttal that this paper generalized the online performative game setting from linear cases to general function cases. And more importantly, the idea of using a function in a certain space to approximate the ground truth offers the possibility to study the behavior of generalization, rather than estimation and optimization. This may be helpful to establish connections between statistical learning and online optimization. The proof in RKHS case exemplifies that we can solve this problem by individually "learning" a time-varying model in some cases. Do I understand correctly? If so, this idea could be made more clearly.
> >
> > For the technical contribution, I keep my point that it is not elaborated properly enough. Even though the proof is not a direct application of (Narang et al,, 2022), it seems to be compositions of several existing techniques. To this point, I am wondering the followings: (i) What is the major difficulty in the online performative game setting studied in this paper? (ii) Which part of the major difficulty is resolved by the technical innovations in this paper (rather than theorems from other papers)? I think a brief answer of these two questions may make the technical contribution of this paper easier to understand.
> >
> > Besides, I noticed some more typos in the current draft, and the authors should have a careful proofreading to correct them.

---

> > > ### Author Response · Authors · 2023-08-21
> > > **Response to Reviewer wDhV (part I)**
> > >
> > > Dear Reviewer,
> > >
> > > Thanks for increasing the score, and for the constructive suggestions provided during the rebuttal and discussion period. We will proofread the current submission and revise typos in the final version of our paper.
> > >
> > > **"In particular, I agree with the saying $\cdots$":**
> > >
> > > The understanding is correct, our approach connects online game optimization with rkhs estimation and can be potentially extended to other problems with the time-varying model. To elaborate, under the parametric model, we form the task of learning the Nash equilibrium of decision-dependent games into a bilevel problem, namely, the lower-level problem is estimating the decision-dependent distribution (or distribution shift), and the upper-level problem uses the estimated model to conduct online gradient descent. By involving the sampling distribution $\rho_{\mathcal{X}}$, we might estimate the parametric model by general function approximation (such as estimating the model by RKHS or even beyond that). Consequently, for other problems with the time-varying model, we might derive an analogous bilevel structure and solve the lower-level problem individually by general function approximation.
> > >
> > > **What is the major difficulty in the online performative game setting studied in this paper?**
> > > The major difficulty is bounding the estimation error for the RKHS case under the power norm $\lVert\cdot\rVert_\gamma$ (instead of the classical RKHS norm $\lVert\cdot\rVert_\mathcal{H}$), namely, Lemma 4 is the most challenging part of our analysis. The distinct properties of the power space $\mathcal{H}^\gamma$ cause the techniques used for the RKHS $\mathcal{H}$ fail to obtain error bounds, as described in **"Technical Challenges of Lemma 4"** in the rebuttal. Briefly speaking, classical theory relies on the fact that the online stochastic approximation (10) is a "contraction map" (i.e. the operator $I-\nu_t(L_t+\lambda_tI)$ is a contraction map on $\mathcal{H}$), but this property does not holds for the power space $\mathcal{H}^\gamma$, this difference further makes the classical methods fail to derive power norm bounds (refer to the ). We will demonstrate the innovation of the approach in the next Section.
> > >
> > > Our motivation for considering the estimation error $f_i^t-f_i$ under the power norm $\lVert\cdot\rVert_\gamma$ is inspired by recent literature on stochastic approximation and kernel method [1][2][3]. Briefly speaking, the concept of power space $\mathcal{H}^\gamma$ and power norm $\lVert\cdot\rVert_\gamma$ bridges the RKHS $\mathcal{H}$ and the L2 space $\mathcal{L}_{\rho_{\mathcal{X}}}^2$ in a continuous scale (lines 265-267). [1][2] finds that we can **choose the power norm properly to accelerate training**, they consider offline spectral filter algorithms and achieve faster convergence rates. Our result (Lemma 4) corresponds with this finding in the online case (refer to **"Interpretation of Results in the RKHS Case"** in the global response for explanations). We remark that techniques for offline algorithms such as the integral operator technique [1][2] do not work in the online case, because the gradient steps are not a constant.

---

> > > > ### Author Response · Authors · 2023-08-21
> > > > **Response to Reviewer wDhV (part II)**
> > > >
> > > > **Which part of the major difficulty is resolved by the technical innovations in this paper**
> > > > To overcome the aforementioned difficulty, we use novel proof steps that are essentially different from the techniques in previous literature about offline algorithms (which typically uses concentration inequalities for the stochastic operator $\sum_{i\in[t]}L_i/t$ to derive error bounds [1][2]).
> > > >
> > > > A major technical innovation is that our analysis decouples power norm bounds by RKHS norm, such methods can be applied to derive power norm bounds for other online algorithms. To elaborate, we decompose $f_t-f$ into three terms $f_{\lambda_t}-f$, $f_{\lambda_t}-g_t$, and $f_t-g_t$ (Lemma 6). While the analysis for $f_{\lambda_t}-f$ (Lemma 7) uses standard spectral decomposition techniques, the proofs for $f_{\lambda_t}-g_t$ and $f_t-g_t$ (Lemma 9 and Lemma 13) are novel. The proofs are based on important observations that $\rVert h\rVert_\gamma = \lVert L_K^{(1-\gamma)/2}h\rVert_{\mathcal{H}}$ for any $h\in\mathcal{H}^\gamma$ and $L_K^{(1-\gamma)/2}$ is **commutative** with the operator $I-\nu_t(L_K+\lambda_t I)$. For instance, in Eq. (122) (line 985), since $L_K^{(1-\gamma)/2}$ is commutative with $\Pi_i^t = \prod_{j = i}^t(I-\nu_j(L_K+\lambda_j I))$, we have $$\lVert\sum_{i = 1}^t\Pi_{i}^t(f_{\lambda_i}-f_{\lambda_{i-1}})\rVert_{\gamma} = \lVert L_K^{(1-\gamma)/2}\sum_{i = 1}^t\Pi_{i}^t(f_{\lambda_i}-f_{\lambda_{i-1}})\rVert_{\mathcal{H}} = \lVert \sum_{i = 1}^t\Pi_{i}^tL_K^{(1-\gamma)/2}(f_{\lambda_i}-f_{\lambda_{i-1}})\rVert_{\mathcal{H}}\leq\sum_{i = 1}^t\lVert \Pi_{i}^t\rVert_{\mathcal{H}\rightarrow\mathcal{H}}\lVert L_K^{(1-\gamma)/2}(f_{\lambda_i}-f_{\lambda_{i-1}})\rVert_{\mathcal{H}}.$$ That is, we leverage the commutativity to decouple the power norm by the operator's RKHS spectral norm $\lVert \Pi_{i}^t\rVert_{\mathcal{H}\rightarrow\mathcal{H}}$ and the RKHS norm $\lVert L_K^{(1-\gamma)/2}(f_{\lambda_i}-f_{\lambda_{i-1}})\rVert_{\mathcal{H}}$. This method is involved in the proofs for Lemmas 9-13 and plays a central role in our analysis.
> > > >
> > > > Another contribution is the recursion decomposition for $f_t-g_t$ (Eq. (86) and page 29), where we decompose $f_t-g_t$ by a sequence of sampling noise iteration ${r_{t}^{(k)}}$ and derive the power norm bound $\lVert r_t^{(k)}\rVert_\gamma$ (Lemma 15). Although an analogous idea was proposed in non-strongly-convex SGD [4], the analysis is essentially difference (because we study this decomposition under the power norm) and we find that the sampling error $f_t-g_t$ can be decomposed as a finite sum of the noise process (Lemma 16), instead of the infinite sum in [4]. This might potentially extend the method to a broader class of problems, particularly in situations where $\lVert r_t^{(k)}\rVert_\gamma$, is not constrained by geometrization sequences.
> > > >
> > > > [1] Lu, Y., Blanchet, J. and Ying, L. (2022). Sobolev acceleration and statistical optimality for learning elliptic equations via gradient descent. arXiv preprint arXiv:2205.07331.
> > > >
> > > > [2] Fischer, S. and Steinwart, I. (2020). Sobolev norm learning rates for regularized least-squares algorithms. The Journal of Machine Learning Research, 21 8464–8501.
> > > >
> > > > [3] Talwai, Prem, Ali Shameli, and David Simchi-Levi. 2022. Sobolev norm learning rates for conditional mean embeddings. In International conference on artificial intelligence and statistics, 10422–10447. PMLR.
> > > >
> > > > [4] Bach, F., & Moulines, E. (2013). Non-strongly-convex smooth stochastic approximation with convergence rate O (1/n). Advances in neural information processing systems, 26.

---

> ### Author Response · Authors · 2023-08-17
>
> Dear reviewer,
>
> Thank you for your insightful review and suggestions. We are wondering if our rebuttal addresses all the concerns. Please feel free to let us know if there are additional questions/concerns.

---

### Official Review · Reviewer_sLZ8 · 2023-07-06

**Soundness:** 3 good
**Presentation:** 3 good
**Contribution:** 2 fair
**Rating:** 7
**Confidence:** 2

**Summary:**

The authors study multi-agent decision-dependent games. Due to strategic interactions among agents, classical gradient-based methods cannot be directly applied in computing Nash equilibrium. Instead, the authors propose using parametric models to model the agents' interactions and the resulting OPGD algorithm. The authors show that, For strongly monotone such games, OPGD converges to NE under (mostly) standard assumptions, including those on the function classes used in defining the parametric models. Numerical experiments on toy game instances complement the theoretical findings.

**Strengths:**

- Simple and implementable algorithms for a new problem setting.
- The convergence rate results use mostly reasonable assumptions that are common in stochastic optimization and online regression.
- Proof sketches are informative and help understand the proof ideas.

**Weaknesses:**

- Strong linearity assumption and simple, 1-agent experiments which seem to be the same as known online regression problems.
-
See **Questions** for details.

**Questions:**

- The experiments are on very simple instances, on which the algorithms seem to degenerate to known online regression algorithms. Are there specific reasons (technical difficulty or otherwise) that the authors decide not to include any experiments with $\geq 2$ agents? It would be helpful to have these experiments and ideally compare with existing algorithms (e.g., Section 4.3 Repeated Stochastic Gradient Method in https://arxiv.org/pdf/2201.03398.pdf). I believe for linear parametric models, OPGD would outperform those as it makes uses of this structure.

- The linearity assumption (Assumption 6) is quite restrictive. However, as a first step toward understanding the convergence behavior of OPGD, this looks ok. It would be valuable to know, via experiments, that the algorithm (Eq. (8) and (9)) still runs and could give high-quality approximate NE even if linearity of the parametric model does not hold. Would you be able to empirically verify this too, by constructing multi-dimensional games such that $f_i(x)$'s are not linear in $x$? More generally, I would try to have new sets of experiments with less structured $f_i$'s, and compare the "linear OPGD", the "kernel function OPGD", and an existing (likely rudimentary) baseline algorithm to demonstrate the benefit of incorporating parametric models.

Regarding the above two questions, for the rebuttal period, it would be sufficient to provide a plan for new experiment setups instead of results. Although it would be great to have preliminary results.

- Line 118: It may help the reader understand the setting further if the authors can add a sentence after the last sentence of this paragraph, stating that (or similar): In round $t$, of the game, each agent $i$ only has access to realized $z_i^{1}, \dots, z_i^{t-1}$, and solves the sampled (aka ERM) version of the optimization formulation (1).

- For (5), if $\mathcal{D}_i(x)$ is a continuous distribution, then exchanging differentiation and integration/expectation requires further (boundedness) assumptions. Please verify if existing assumptions are sufficient and point this out.
- In Theorem 1, $D_1$ involves $t$ and should *not* be a constant? Maybe $t$ should be $1$ instead (since $t\geq 1$)?

**Limitations:**

The authors have stated the limitations in terms of technical assumptions. I do not see any potential negative societal impact.

---

> ### Author Rebuttal · Authors · 2023-08-09
>
> Thanks for your insightful comments and kind suggestions! We clarify our novelty over Narang et al. (2022) in the global response. We refer the reviewer to **Novel I** for the comparison of OPGD and Algorithm 1 and Narang et al. (2022). We refer to **Novel II** for the explanation that extending the linear algorithm into the non-linear (RKHS) case is not trivial and our approach (introduce the dynamic regularization term $\lambda_t$). We summarize our contribution to the convergence analysis under the power norm in **Novel III** and present a detailed explanation for Lemma 4 and Theorem 2 in **Interpretation of Results in the RKHS Case**. In the following, we provide responses to your questions.
>
> **"Strong linearity assumption and simple $\cdots$":**
> We propose OPGD for both the linear and kernel function classes. In fact, linear OPGD (Algorithm 1) depends on the linear assumption (Assumption 6) and Assumption 7, and the kernel OPGD (Algorithm 2) depends on Assumption 8, 9, 10 and does not require Assumptions 6 or 7. In more detail, the parametric assumption of kernel OPGD is Assumption 8, which assumes $f_i\in(\mathcal{H}^\beta)^p$. The power space $\mathcal{H}^\beta$ is non-linear, thus, the kernel OPGD can address the non-linear parametric model. Besides, we can potentially extend the kernel OPGD to larger function classes, such as overparameterized neural networks using the technique of neural tangent kernel [1].
>
> **"The experiments are on very simple instances $\cdots$":**
> We present the supplemental experiments on multi-agent decision-dependent games (2-agent games) in **supplemental_experiment.pdf**. Figure 1 shows that the linear OPGD matches the theoretical rate $\mathcal{O}(t^{-1})$. Figure 2 compares the linear OPGD with Repeated Stochastic Gradient Method (Narang (2022) Section 4.3, which is the baseline algorithm in performative prediction to find the stable equilibrium) and Adaptive Gradient Method (Narang (2022) Algorithm 1). Figure 2 shows that Linear OPGD has the same convergence rate with Adaptive Gradient Method, and Repeated Stochastic Gradient Method fails to find the Nash equilibrium because it does not leverage the parametric model.
>
>
>
> **"The linearity assumption (Assumption 6) is quite restrictive $\cdots$":**
> The kernel OPGD is capable of finding the Nash equilibrium in non-linear models. We present the result of a 2-agent decision-dependent game with the non-linear parametric model in **supplemental_experiment.pdf**. Figure 3 shows that the kernel OPGD matches the theoretical rate $\mathcal{O}(t^{-1/2})$. Figure 4 compares the kernel OPGD with Linear OPGD and Adaptive Gradient Method. Figure 4 shows that kernel OPGD converges to the Nash equilibrium as our theory predicts, but Linear OPGD and Adaptive Gradient Method fail to find NE.
>
> **"Line 118: It may help the $\cdots$":**
> Thanks for your helpful advice! In the round $t$, the agent $i$ only has access to $z_i^1,\cdots,z_i^{t-1}$ and $x^1,\cdots,x^{t-1}$. We will explain the oracle of agents in this paragraph for better understanding.
>
> **"For (5), if $\mathcal{D}_i(x)$ is $\cdots$":**
> We view interchangeability as a basic assumption in our paper (we will clarify this in the revision). Here we derive a sufficient condition for this assumption. Under the parametric assumption, $\mathcal{L}_i(x)$ can be rewritten as
>
> $\mathbb{E}_{\epsilon_i\sim\mathcal{P}_i} \ell_i(x,f_i(x)+\epsilon_i)$ (line 156). Since we have assumed $\ell_i,f_i$ are continuous and have continuous partial derivatives, then to ensure the differentiation and expectation are interchangeable, we need additional assumptions such as $\ell_i(x,f_i(x)+\epsilon_i)$ is integrable with respect to the measure $\epsilon_i\sim\mathcal{P}_i$ for any $x\in\mathcal{X}$ and the improper integral $\int\partial\ell_i(x,f_i(x)+\epsilon_i)/\partial x_i~d\mathcal{P}_i$ converges uniformly with respect to $x$. As a result, a sufficient condition is: $\ell_i$ is Lipschitz continuous in $x$ and $z_i$, and $f_i$ is Lipschitz in $x$, which is the same sufficient condition for Assumption 4 and Assumption 5.
>
> **"In Theorem 1, $\cdots$":**
> We apologize for the typo, the true expression is $D_1 := 4\zeta^2(1+2(M/(t_0+1)+\sup_{i\in[n]}\lVert A_i\rVert_F^2))$, where $D_1$ is a constant and not related to iteration $t$. We will correct this typo in the revision.
>
> [1] Allen-Zhu, Z., Li, Y., & Liang, Y. Learning and generalization in overparameterized neural networks, going beyond two layers. NIPS, 2019.

---

> > ### Comment · Reviewer_sLZ8 · 2023-08-18
> >
> > Thank you for the detailed comments. The additional experiments look good to me. Please make sure to add brief descriptions of the findings in the experiments, specifically for settings where baseline/simpler algorithms fail (e.g., Figure 2 and Figure 4).
> >
> > I may follow up on the novelty claims, but for now, they look ok to me. In particular, I agree that II and III involve non-trivial technical developments. Experiment results may help strengthen their significance, that is, why one would be interested in studying/applying Kernel OPGD over simpler algorithms.

---

> > > ### Author Response · Authors · 2023-08-21
> > >
> > > Dear Reviewer,
> > >
> > > Thanks for your thoughtful comments. We will include descriptions of settings and findings of the experiment comparing OPGD and baseline algorithms in the final version of our paper.
> > >
> > > For Figure 2, the two-player decision-dependent game with the linear parametric model is constructed as Section 1.1 **supplemental_experiment.pdf** describes. We compare the linear OPGD with adaptive gradient methods (AGM, Narang (2022) Algorithm 1) and Repeated Stochastic Gradient Method (RSGM, Narang (2022) Section 4.3). AGM estimates the decision-dependent distribution by the method described in the global response **Novelty I**. RSGM uses the first term of the performative gradient (4) for gradient descent, namely, it cannot characterize the decision-dependent distribution. Letting $t_0 = 10$, for the linear OPGD, we set the gradient steps as $\eta_t = 6/(t+t_0),\nu_t = 6/(t+t_0)$. For AGM, we set the injective noise as $\mathcal{N}(0,0.32)$ and the gradient steps are the same as linear OPGD. For RSGM, we set the gradient steps as $\eta_t = 5/(t+t_0)$. Figure 2 shows that the result for these three methods averaged 20 different seeds. For the decision-dependent game with linear parametric function, linear OPGD and AGM converge to the Nash equilibrium $(0.5,1)$ with the same rate $\mathcal{O}(1/t)$, while RSGM fails to find the Nash equilibrium. This is because RSGM only uses the term $\nabla_i\ell_i(x,z_i)$ for the gradient descent, and ignores the dependence between the distribution $\mathcal{D}_i$ and the joint action $x$ (Section 4, Narang (2022)).
> > >
> > > For Figure 4, the construction of the game is described in Section 1.2. We compare the Kernel OPGD with the linear OPGD and AGM. We set the gradient steps of linear OPGD and AGM as $\eta_t = 4/(t+t_0),\nu_t = 4/(t+t_0)$ where $t_0 = 10$, and set the injective noise of AGM as $\mathcal{N}(0,0.32)$. The gradient steps and regularization terms of the kernel OPGD are $\eta_t = 6/(t+t_0),\nu_t = 7/(t+t_0)^{3/4},\lambda_t = 7/(t+t_0)^{1/4}$. Figure 4 shows that for proposed the non-linear parametric function, the kernel OPGD converges to the Nash equilibrium $(0.5, 0.5)$, while both the linear OPGD and AGM fail to find the NE. In fact, the linear OPGD and AGM approximate the parametric function by linear model and have large estimation error, thus, the estimated performative gradient (6) is biased and makes the projected gradient descent fail to converge to NE.

---

> ### Author Response · Authors · 2023-08-17
>
> Dear reviewer,
>
> Thank you for your insightful review and suggestions. We are wondering if our rebuttal addresses all the concerns. Please feel free to let us know if there are additional questions/concerns.

---

### Official Review · Reviewer_QMrd · 2023-07-07

**Soundness:** 3 good
**Presentation:** 4 excellent
**Contribution:** 2 fair
**Rating:** 5
**Confidence:** 3

**Summary:**

This work proposes stochastic approximation-based Online Performative Gradient Descent to compute the Nash equilibrium of multi-agent decision-dependent games with bandit feedback. To this end they extend the work of Narang et al. (2022) in two ways: 1. The work uses stochastic approximation method to estimate the best linear operator $A_i$ to approximate the distribution map whereas Narang et al. (2022) assumes $A_i$ to be known. 2. Extend the result to the RKHS setting.

**Strengths:**

1. The work studies a relevant and interesting problem.

2. The experimental results are promising.

3. Extension of results from Narang et al. (2022) to RKHS extend the applicability of the results.

**Weaknesses:**

The theoretical novelty compared to Narang et al. (2022) seems inadequate and flawed to some extent.

1.I am concerned about the use of $u_i^t$ to approximate $A_i$. The main challenge of problems with decision-dependent distribution is that the distribution depends on the *decisions* that is $x_i^t$ which makes estimation of $A_i$ difficult. OPGD assumes access to random samples from the distribution $\rho_X$, and observes corresponding $y$ and use that to estimate $A_i$. If one assumes access to a separate sample $(u_i,y_i)\sim \rho_i$ then estimating $A_i$ is a simple stochastic approximation procedure and the corresponding theory is well-known in the online SGD and stochastic approximation (eqn (27), (28)). Combining these results with the ones (Theorem 6) in Narang et al. (2022) directly gives Theorem 1.

2. Say it was indeed possible to query such a point $u_i$. Then how to estimate $A_i$ is already explored in Section 6.3 of Narang et al. (2022) and it's almost the same method as OPGD with same theoretical rate (Theorem 7).

3. The extension to RKHS increases the applicability but the proof techniques are almost the same, albeit involve more tedious calculations, as the linear case which is expected since it is after all a linear setting. The authors concur as well (line 310).

4. Minor point: Assumption 7 is satisfied by Gaussian distn but here $\rho_X$ is supposed to have a compact support since $X$ is compact. So $\rho_X$ can't be Gaussian. But anyway, Assumption 7 can be satisfied by other distribution and it's not a major concern.

**Questions:**

If you could clarify your novelties over Narang et al. (2022), I would love to reconsider my score.

---

> ### Author Rebuttal · Authors · 2023-08-09
>
> Thanks for your detailed and valuable review. We have outlined the contribution of our work in the global comment. Next, we will address the questions not included in that particular response.
>
> **"I am concerned about the use of $\cdots$":**
> Thanks for this question. We regard the use of $u_i^t$ as proper. Intuitively, an agent taking the action $u_i^t$ can be viewed as an exploration of the action space. In more detail, to learn the strategic behavior of other agents, it is feasible for agents to explore the space under certain personalized rules, namely, taking actions under the user-specified sampling distribution $\rho_{\mathcal{X}}$. A major benefit of considering $u_i^t$ and $\rho_{\mathcal{X}}$ is that we can extend the algorithm to the non-linear function class,  while the method in Narang et al. (2022) fails, we will explain this in the next answer. As a result, our learning framework can be generalized to larger function classes such as RKHS or even beyond that, such as overparameterized neural networks using the technique of neural tangent kernel.
>
> **"Say it was indeed possible to query $\cdots$":**
> We remark that our method is essentially different from Narang et al. (2022). Since the linear assumption is restrictive, a major motivation of our work is learning decision-dependent games with a non-linear parametric model, which is more applicable. However, the method in Narang et al. (2022) fails for the non-linear parametric model. To elaborate, we briefly summarize the sampling strategy in Narang et al. (2022) Algorithm 1: In iteration $t$, sample a $z_i^t$ at current the action $z_i^t\sim\mathcal{D}_i(x^t)$ and let the agent play again with an inject noise $u^t$ to obtain a sample $q_i^t\sim\mathcal{D}_i(x^t+u^t)$.
>
> Narang et al. (2022) leverages the difference between $z_i^t$ and $q_i^t$ to estimate the unknown parametric matrix $A_i$, their approach is based on the fact that $\mathbb{E}[q_i^t-z_i^t|u^t,x^t] = A_iu^t$. Since $\mathbb{E}[q_i^t-z_i^t|u^t,x^t]$ is not related with $x^t$, $A_i$ can be estimated by ordinary least square. We remark that this approach depends on the linear parametric assumption and cannot be generalized to the non-linear case. In more detail, consider this sample strategy in the RKHS case, we have $\mathbb{E}[q_i^t-z_i^t|u^t,x^t] = f_i(x^t+u^t)-f_i(x^t) = \langle f_i,\phi_{x^t+u^t}-\phi_{x^t}\rangle_{\mathcal{H}}$. Thus, $\mathbb{E}[q_i^t-z_i^t|u^t,x^t]$ depended on agents' actions, and the estimation of $f_i$ is related to the dynamics of $\{x^t\}$ (approximate $f_i$ not a simple non-parametric regression because the pattern of $\{x^t\}$ is influenced by the project gradient descent). Intuitively, the change of agents' actions in different iterations brings additional errors and makes the estimation difficult and even fails to converge.
>
> **"The extension to RKHS increases the applicability $\cdots$":**
> We apologize for the confusion in line 310: "The proof is similar to that of Theorem 1." Due to the limitation of pages, we omit the sketch of proof for Theorem 2 and here we aim to convey that the structure of proof for Theorem 2 is similar to the linear case, namely, Theorem 2 is guaranteed by the estimation error (Lemma 4) and the one-step error (Lemma 5). However, bounding the estimation error (Lemma 4) is not trival and we acknowledge this key lemma as a major contribution of our paper. Briefly speaking, we study the estimation error of the online stochastic approximation (10) under the power norm $\lVert\cdot\rVert_\gamma$ and obtain a fast rate $\mathcal{O}(t^{-(\beta-\gamma)/(\beta-\gamma+2)})$, in contrast to the classical minimax optimal rate of one-pass SGD $\mathcal{O}(t^{-(\beta-1)/(\beta+1)})$ under the RKHS norm $\lVert\cdot\rVert_{\mathcal{H}}$.
>
> We remark the standard techniques of online stochastic approximation under the RKHS norm fail in the power norm, the main difficulty arises from the differing properties between the power space $\mathcal{H}^\gamma$ and the RKHS $\mathcal{H}$. For the online estimation step (75) (we will use the notation in Appendix E in the following text), the standard methods depend on the fact that $I-\nu_t(L_t+\lambda_tI)$ is a contraction map on $\mathcal{H}$ (because the sampling operator $L_t$ is compact, self-adjoint, and positive-definite on this space). However, for the power space $\mathcal{H}^\gamma$, $L_t$ is not self-adjoint or positive-definite. To obtain the error bound under the power norm, we take a series of novel proof steps, such as considering the semi-stochastic population iteration $g_t$ (76) and applying the semi-stochastic decomposition recursively (86) to decompose the semi-stochastic sampling noise $g_t-f_t$. We refer the reviewer to Appendix E (Specifically, Page 26) for the proof outline.
>
> **"Minor point: Assumption 7 is satisfied by $\cdots$":**
> Thanks for pointing out the mistakes! We will revise this example in the revision. In fact, we use $\rho_{\mathcal{X}} = \mathcal{U}[0,1]$ in the numerical experiments, which satisfies the assumption.
>
> **"If you could clarify your novelties over Narang et al. (2022) $\cdots$":**
> We present the novelties in the global response. We refer the reviewer to **Novel I** for the comparison of OPGD and Algorithm 1 and Narang et al. (2022). We refer to **Novel II** for the explanation that extending the linear algorithm into the non-linear (RKHS) case is not trivial and our approach (introduce the dynamic regularization term $\lambda_t$). We summarize our contribution to the convergence analysis under the power norm in **Novel III** and present a detailed explanation for Lemma 4 and Theorem 2 in **Interpretation of Results in the RKHS Case**. Also, we present addition experiments on multi-agent decision-dependent games, and compare linear OPGD, kernel OPGD, and AGM (Algorithm 1 in Narang et al. (2022)) in **supplemental_experment.pdf**.

---

> ### Author Response · Authors · 2023-08-17
>
> Dear reviewer,
>
> Thank you for your insightful review and suggestions. We are wondering if our rebuttal addresses all the concerns. Please feel free to let us know if there are additional questions/concerns.

---

> ### Comment · Reviewer_QMrd · 2023-08-20
> **Thanks for the responses.**
>
> I am happy with the authors' response. I have increased my score.

---

> > ### Author Response · Authors · 2023-08-21
> >
> > Dear Reviewer,
> >
> > Thanks for raising the score, and for the thoughtful consideration during the rebuttal and discussion period. We will add a summary to demonstrate the novelty in the final version of our paper.

---

### Official Review · Reviewer_jNzm · 2023-07-07

**Soundness:** 3 good
**Presentation:** 3 good
**Contribution:** 3 good
**Rating:** 8
**Confidence:** 3

**Summary:**

This work studies the framework of decision dependent games, where the actions of the agents in the game do not only affect the utilities of the players but also the distribution of population data observed by them. The critical challenge of this type of games is that the data population sampling process is not in general differentiable with respect to the parameters of the agents. The authors circumvent this by proposing a noisy stochastic differentiable model of the sampling distribution. The model itself is parametric and the authors propose OPGD an algorithm that learns both a the model parameters and the players strategy. The authors analyze both the linear and the RKHS setting for strongly monotone games.

**Strengths:**

The authors have clearly explained the proposed framework and the necessary background to understand the challenges involved. While elements of the linear model setting existed in Narang et al, at least the RKHS component is to the best of my knowledge novel. The experimental results may be helpful to practitioners and may lead to practical (in addition to the theoretical) impact.

**Weaknesses:**

I think the differentiation between Narang et al and this work should be made more clear throughout. More specifically, what is the unique insight that allowed this work to arrive at the novel results?

**Questions:**

Is it possible for the players to first approximately learn the parameters of $f$ through the bandit feedback and then armed with that knowledge play the monotone game? If this is the case, how do the proposed algorithms compare to this baseline?

**Limitations:**

Limitations are addressed by the authors.

---

> ### Author Rebuttal · Authors · 2023-08-09
>
> Thanks for your positive feedback and the time spent reviewing and understanding our paper! We have added more explanation on the contribution and innovation of our work in the global response. We address the other comments as follows:
>
> **"I think the differentiation between Narang et al $\cdots$":**
> We have highlighted our contribution over Narang et al. (2022) in the overall response. We refer the reviewer to **Novelty I** for our innovation of the parametric model and a statement of the difference between OPGD and Algorithm 1 in Narang et al. (2022). We refer to **Novelty II** for an explanation of the difficulty in the RKHS case and our approach. We refer to **Novelty III** for our theoretical contribution of the convergence analysis under the power norm, we interpret Lemma 4 and Theorem 2 in **Interpretation of Results in the RKHS Case** for better understanding.
>
> In addition to the aforementioned contributions, we briefly demonstrate the technical challenge of bounding the estimation error under the power norm (Lemma 4) and our approach. Generally speaking, the standard method for convergence analysis under the RKHS norm fails in bounding the power norm [1]. Intuitively, the main difficulty arises from the differing properties between the power space $\mathcal{H}^\gamma$ and the RKHS $\mathcal{H}$. For the online estimation step (75) (we will use the notation in Appendix E in the following text), the standard methods depend on the fact that $I-\nu_t(L_t+\lambda_tI)$ is a contraction map on $\mathcal{H}$ (because the sampling operator $L_t$ is compact, self-adjoint, and positive-definite on this space). However, for the power space $\mathcal{H}^\gamma$, $L_t$ is not self-adjoint or positive-definite. Because for any $h_1,h_2\in\mathcal{H}^\gamma$, by the definition of $L_t$ (here we lift the domain from $\mathcal{H}$ to $\mathcal{H}^\gamma$), we have $L_t(h_i) = h_i(x^t)\phi_t = \langle h_i,\phi_t^\gamma\rangle_\gamma\phi_t$ and $\langle L_th_1,h_2\rangle_\gamma\neq\langle h_1,L_th_2\rangle_\gamma$.
>
> To handle this problem, we use novel proof steps and decompose $L_t$ by the covariance operator $L_K$ (which is the mean operator of $L_t$) and $L_t-L_K$. To elaborate, we consider the semi-stochastic population iteration $g_t$ (76) and decompose $f_t-f$ by $f_{\lambda_{t}}-f$, $g_t-f_{\lambda_{t}}$, and $f_t-g_t$, where $f_{\lambda_t}$ is the solution of the kernel ridge regression (line 762). Compared with $f_t$ (75), iterations of $g_t$ replace the stochastic operator $L_t$ with $L_K$ and remain the random term $y_t\phi_t$.
>
> Intuitively, $f_{\lambda_t}\rightarrow f$ if $\lambda_t\rightarrow 0$, because $f$ is the minimizer of the unregularized regression (7). We obtain the power norm bound for $f_{\lambda_t}-f$ by spectral decomposition (Lemma 7).
>
> To analyze $g_t-f_{\lambda_{t}}$, we derive the martingale decomposition (81) and observe that $L_K$ is commutative with the operator $\Pi_i^j$. This allows us to use Lemma 18 ($\lVert f_{\lambda_t}-g_t\rVert_\gamma = \lVert L_K^{(1-\gamma)/2}(f_{\lambda_t}-g_t)\rVert_{\mathcal{H}}$) to transfer the analysis of power norm bound to RKHS norm bound (Lemma 9).
>
> For the term $f_t-g_t$, which is mainly characterized by the sampling error operator $L_K-L_t$, we use semi-stochastic decomposition (86) recursively to decompose this term. To elaborate, we observe that $f_t-g_t$ has the same iteration structure as $f_t$ and thus consider its semi-stochastic population iteration $r_t^{(0)}$. We further consider the error $f_t-g_t-r_t^{(0)}$ and use semi-stochastic decomposition recursively to define a noise process $\{r_{t}^{(k)}\}$ (86). We prove that $f_t-g_t$ can be written as a finite sum of $\{r_{t}^{(k)}\}$ (Lemma 16) and obtain the power norm bound of $r_{t}^{(k)}$ (Lemma 14).
>
> **"Is it possible for the players to first approximately learn $\cdots$":**
> To begin with, we remark that agents in decision-dependent games are very likely to face an online decision process: when the agents take action, the distribution of observed data changes. Thus, we want to play and learn the game simultaneously. Besides, it is indeed possible that we first learn the parametric model by offline data and plug it in to find the Nash equilibrium. In this case, $f_i$ can be estimated by the sampled version of the non-parametric regression (7) and the sample efficiency of the data-collection process might be an important concern. As a result, it may be helpful to study the optimal design to derive efficient offline algorithms. Generally speaking, the comparison of sample complexity and computational cost between offline and online algorithms depends on particular data-generation assumptions and would be a future direction for research.
>
> [1] Tarres, P. and Yao, Y. Online learning as stochastic approximation of regularization paths: Optimality and almost-sure convergence. IEEE Transactions on Information Theory, 2014.

---

> ### Author Response · Authors · 2023-08-17
>
> Dear reviewer,
>
> Thank you for your insightful review and suggestions. We are wondering if our rebuttal addresses all the concerns. Please feel free to let us know if there are additional questions/concerns.

---

> > ### Comment · Reviewer_jNzm · 2023-08-19
> > **All my questions were answered**
> >
> > All of my questions are answered. I have decided to increase my score. It would be great to add a summary of the novelty discussions in the paper as well.

---

> > > ### Author Response · Authors · 2023-08-21
> > >
> > > Dear Reviewer,
> > >
> > > Thanks for increasing the score, and for the constructive suggestions provided during the rebuttal and discussion period. We will add a summary of the novelty discussions in the final version of our paper.

---

### Official Review · Reviewer_AyUq · 2023-07-13

**Soundness:** 4 excellent
**Presentation:** 4 excellent
**Contribution:** 2 fair
**Rating:** 5
**Confidence:** 4

**Summary:**

The authors focus on the analysis of learning the Nash equilibrium (which is unique) in strongly monotone games that are decision-dependent. They provide a definition of decision-dependent games generalizing a recent work of Narang et al. Intuitively speaking, if an agent chooses an action, he gets an observation (that depends on a distribution that is parametrized of everyone's actions) and his utility depends on the observed data (bandit setting) and everyone's actions. The authors provide an algorithm that converges to the Nash equilibrium in a rate of 1/t when the observed data depend linearly in the actions (linear regression type dependent-game), i.e., the focus is linear function classes. Further they focus on function kernel classes and provide similar type rates on convergence to the Nash equilibrium.

**Strengths:**

The idea of decision-dependent games is interesting, though it was firstly defined in another paper.

**Weaknesses:**

The algorithm (projection type gradient descent with also updating the estimate on the A matrix) and the analysis seems not novel and not surprising. I think the paper would be more interesting if the authors could argue for monotone and not only strongly monotone games or even going beyond that (e.g., games that satisfy MVI for example).

**Questions:**

What is the main technical challenge of your work?

---

> ### Author Rebuttal · Authors · 2023-08-09
>
> Thanks for your time and effort in reviewing our paper. We have summarized the contribution and innovation of our paper over Narang et al. (2022) in the global response. In the following, we address other comments not covered in that response.
>
> **"I think the paper would be more interesting $\cdots$":**
> Thanks for this insightful question. We acknowledge that our work can be used to solve general monotone game or even games with weak MVI condition. The reason is our method features estimation of the performative gradient, which can be incorporated into other first-order methods for solving games [1]. We consider the strongly monotone games as an initial step for studying decision-dependent games. Extending to a more general class of decision-dependent games will be an important future direction.
>
> **"What is the main technical challenge of your work?":**
> The technical challenges lies in three aspects: 1. we propose a general learning framework for decision-dependent game; 2. we introduce a dynamic regularization term $\lambda_t$ in the online estimation to insure convergence; 3. we bound the estimation error of online stochastic approximation (10) under the power norm (Lemma 4). We refer the reviewer to **Novel I**  and **Novel II** in the global response for the demonstration on technical challenges of 1 and 2. We have summarized our result for 3 in **Novel III** and **Interpretation of Results in the RKHS Case**, now we provide a detailed explanation for the technical challenge of Lemma 4.
>
> **Technical Challenges of Lemma 4.** In the following, we use the notation defined in Appendix E. Intuitively, the main challenge to derive power norm bounds for iteration (75) (i.e. $f_t$) under the norm $\lVert\cdot\rVert_\gamma$ arises from the differing properties between the power space $\mathcal{H}^\gamma$ and the RKHS $\mathcal{H}$. To elaborate, the standard method to derive error bounds under $\lVert\cdot\rVert_\mathcal{H}$ decomposes the error $f-f_t$ based on the operator $I-\nu_t(L_t+\lambda_tI)$ (such as Theorem 3.1 in [2]), and the analysis is based on the fact that $I-\nu_t(L_t+\lambda_tI)$ is a contraction map on $\mathcal{H}$. This is because the sampling operator $L_t$ is compact, self-adjoint, and positive-semidefinite on $\mathcal{H}$, thus, the spectral theorem implies $\lVert I-\nu_t(L_t+\lambda_tI)\rVert_{\mathcal{H}\rightarrow\mathcal{H}}\leq 1-\nu_k\lambda_k$. However, this operator does not exhibit the same behavior on the power space $\mathcal{H}^\gamma$. By the definition that $L_t=\phi_t^*\phi_t$, for any $h_1,h_2\in\mathcal{H}^\gamma$, $L_t(h_i) = h_i(x^t)\phi_t=\langle h_i,\phi_t^\gamma\rangle_\gamma\phi_t$ and $\langle L_th_1,h_2\rangle_\gamma\neq\langle h_1,L_th_2\rangle_\gamma$ (here we lift the domain of $L_t$ from $\mathcal{H}$ to $\mathcal{H}^\gamma$). Thus, $L_t$ is not self-adjoint or positive-definite on $\mathcal{H}^\gamma$ and the spectral norm $\lVert I-\nu_t(L_t+\lambda_tI)\rVert_{\mathcal{H}^\gamma\rightarrow\mathcal{H}^\gamma}$ might larger than 1.
>
> **Our Approach.** To overcome the aforementioned obstacle, our approach has four steps:
> 1. Since $L_t$ is stochastic and hard to analyze under the power norm, we consider the covariance operator $L_K$ and decompose the error $f-f_t$ by $L_K$ and $L_K-L_t$ ($L_K$ is the mean operator of $L_t$). We define the semi-stochastic population iteration (76) (i.e. $g_t$), which replaces $L_t$ with $L_K$ and remains the stochastic term $y_t\phi_t$. We further decompose the error $f-f_t$ (Lemma 6 and Remark 2) and aim to derive the power norm bounds of $f-f_{\lambda_t}$, $f_{\lambda_t}-g_t$, and $f_t-g_t$.
> 2. For the term $\lVert f-f_{\lambda_t}\rVert_\gamma$, we obtain the error bound by spectral decomposition (Lemma 7). We remark that $f_{\lambda_t}$ is the solution of the kernel ridge regression with regularization $\lambda_t$ (line 763) and $f$ minimizes the unregularized kernel regression. Intuitively, $\lVert f-f_{\lambda_t}\rVert_\gamma$ converges to zero if $\lambda_f\rightarrow0$.
> 3. For the term $\lVert f_{\lambda_t}-g_t\rVert_\gamma$, we leverage $L_K$ to bridge the analysis between power norm and RKHS norm. Our estimation (Lemma 9) is based on the observation that $L_K$ is commutative with the operator $\Pi_i^j$ (defined in (81)). This crucial property allows us to combine the martingale decomposition (81) with the fact $\lVert f_{\lambda_t}-g_t\rVert_\gamma = \lVert L_K^{(1-\gamma)/2}(f_{\lambda_t}-g_t)\rVert_{\mathcal{H}}$ (Lemma 18) and derive the error bound under the RKHS norm.
> 4. For the term $\lVert f_t-g_t\rVert_\gamma$, we decompose $f_t-g_t$ by applying semi-stochastic decomposition (86) recursively. We observe that the iteration of $f_t-g_t$ (87) has a similar structure with $f_t$ (76), thus, we repeat the method in step 1 and consider the semi-stochastic population iteration of $f_t-g_t$, namely $r_t^{(0)}$ (line 842). We find that $f_t-g_t-r_t^{(0)}$ also has the same iteration structure as $f_t$ and $f_t-g_t$ (line 844). As a result, we repeat this procedure and define a sequence of noise process $\{r_t^{(k)}\}$ (Intuitively, we decompose $L_K-L_t$ by this recursive decomposition. See line 837 and more explanation in Appendix F6.2) and prove that $f_t-g_t$ can be written as a finite sum of noise processes (Lemma 16). Moreover, since each noise process is updated by $r_t^{(k)} = (I-\nu_t(L_K+\lambda_t I))r_{t-1}^{(k)}+\nu_t(L_K-L_t)r_{t-1}^{(k-1)}$, we leverage the commutability between $L_K$ and $(I-\nu_t(L_K+\lambda_t I))$ and use the same technique in step 3 to obtain the power norm bound of each noise process(Lemma 14).
>
> [1] Pethick, T., Fercoq, O., Latafat, P., Patrinos, P., & Cevher, V. Solving stochastic weak Minty variational inequalities without increasing batch size. arXiv preprint arXiv:2302.09029,2023.
>
> [2] Tarres, P. and Yao, Y. Online learning as stochastic approximation of regularization paths: Optimality and almost-sure convergence. IEEE Transactions on Information Theory,2014.

---

> ### Author Response · Authors · 2023-08-17
>
> Dear reviewer,
>
> Thank you for your insightful review and suggestions. We are wondering if our rebuttal addresses all the concerns. Please feel free to let us know if there are additional questions/concerns.

---

> > ### Comment · Reviewer_AyUq · 2023-08-18
> >
> > Thank you for your response. It seems that RKHS was quite demanding so I decided to increase my score. However, it still feels that Narang et al paper takes away some of the novelty of this submission.

---

> > > ### Author Response · Authors · 2023-08-21
> > >
> > > Dear Reviewer,
> > >
> > > Thanks for raising the score. Your thoughtful comments and insightful feedback provided during the rebuttal and discussion period have been invaluable in enhancing the quality of our work.

---

### Author Rebuttal · Authors · 2023-08-09

In the following, we present a detailed explanation of our novelty over Narang et al. (2022).

**Novelty I.** We extend the linear model (Assumption 6) in Narang et al. (2022)  (which assumes agents share a common parametric matrix $A = [A_i~A_{-i}]$) into a general framework (parametric functions $f_i$ are different among agents). We derive sufficient conditions for the game to be strongly monotone (Proposition 2). We propose OPGD for the linear function class and extend it to the kernel function class (the extension is non-trivial and we refer to **Novelty II** for explanations).

The core problem in decision-dependent games is estimating the performative gradient, and OPGD uses a different method for this compared with Narang et al. (2022). To elaborate, Algorithm 1 in Narang et al. (2022) samples $z_i^t\sim\mathcal{D}_i(x^t)$ at current the action and let agents play again with an injected noise $u^t$ to obtain $q_i^t\sim\mathcal{D}_i(x^t+u^t)$. Their approach is based on the fact that $\mathbb{E}[q_i^t-z_i^t|u^t,x^t]=A_iu^t$,which is not related to $x^t$. Thus, $A_i$ can be estimated by online least square.

However, this method cannot be generalized to non-linear (RKHS) cases, because $\mathbb{E}[q_i^t-z_i^t|u^t,x^t]=f_i(x^t+u^t)-f_i(x^t)=\langle f_i,\phi_{x^t+u^t}-\phi_{x^t}\rangle_{\mathcal{H}}$. Thus, $\mathbb{E}[q_i^t-z_i^t|u^t,x^t]$ depends on agents' actions and the change of action will bring additional error that makes the estimation fail to converge. Besides, there is a minor issue that $x^t+u^t$ might go out of the compact action space.

OPGD lets agents play $u_i^t\sim\rho_{\mathcal{X}}$ to explore the action space and learn the strategic behavior of other agents. OPGD estimates the parametric function by solving the ERM version of (7) using online stochastic approximation (8), (10). This learning framework can be extended to RKHS and potentially beyond that, such as overparameterized neural networks using the technique of neural tangent kernel [1].

**Novelty II.** The major difficulty in the RKHS case is that an RKHS generally has infinite dimensions, and solving the ERM version of (7) leads to ill-posed solutions. Mathematically, by Mercer's theorem, the covariance operator $L_K$ has the spectral decomposition (23) and its eigenvalues satisfy $\mu_i\rightarrow0$. Thus, $L_K-\ell_1I$ is not a positive operator for any $\ell_1>0$ and $L_K\succeq\ell_1I$ does not hold. As a result, Assumption 7 is not satisfied in the RKHS case and $u_i^t\sim\rho_{\mathcal{X}}$ cannot sufficiently explore the action space.

To handle this problem, we introduce the regularization term $\lambda_t$ and aim to solve a kernel ridge regression (line 190). Since the kernel ridge has a biased solution $f_{\lambda_t}$, we let $\lambda_t$ shrink to $0$ gradually to ensure $f_{\lambda_t}\rightarrow f$. We remark that the change of $\lambda_t$ will bring drift error $f_{\lambda_t}-f_{\lambda_{t-1}}$, which is closely nested with the estimation error $f_t-f$. We choose $\nu_t$ and $\lambda_t$ carefully to let $f_{\lambda_t}-f$ and $f_t-f_{\lambda_t}$ converge simultaneously (Lemma 4).

**Novelty III.** We study the convergence rate of the online stochastic approximation for linear function class (Lemma 1) and kernel function class (Lemma 4) and our technical contribution mainly lies in the kernel case. We derive the first convergence analysis for online stochastic approximation under the power norm $\lVert\cdot\rVert_{\gamma}$ and obtain a fast rate $\mathcal{O}(t^{-(\beta-\gamma)/(\beta-\gamma+2)})$ (Lemma 4), in contrast to the classical minimax optimal rate for one-pass SGD under the RKHS norm $\mathcal{O}(t^{-(\beta-1)/(\beta+1)})$ [2][3]. To elaborate, we leverage the embedding property (Assumption 9) to show that one can choose the power norm properly to achieve faster convergence rates (we reiterate our results in the next paragraph). To obtain the estimation error bounds under the power norm (Lemma 4), we use novel proof steps such as considering the semi-stochastic population iteration $g_t$ (76), deriving the martingale decomposition (81), and decomposing the semi-stochastic sampling noise $g_t-f_t$ by applying the semi-stochastic decomposition recursively (86).

**Interpretation of Results in the RKHS Case.** Lemma 4 presents the error bound of online stochastic approximation (10) under the power norm $\lVert\cdot\rVert_\gamma$, our result includes the classical theory under the RKHS norm $\lVert\cdot\rVert_{\mathcal{H}}$ and extends it in a continuous scale. In more detail, for any $\gamma\in[\alpha,\beta)$ and $\gamma\leq1$, Lemma 4 describes how to choose the step-sizes $\nu_t$ and regularization term $\lambda_t$ properly (line 598) to insure convergence under $\lVert\cdot\rVert_\gamma$ with the convergence rate $\mathcal{O}(t^{-(\beta-\gamma)/(\beta-\gamma+2)})$. For $\beta>1$ and $\gamma=1$, this rate would be $\mathcal{O}(t^{-(\beta-1)/(\beta+1)})$ and matches the optimal rate under the RKHS norm. If the embedding property (Assumption 9) holds for some $\alpha<1$, we choose $\gamma=\alpha$ to achieve a faster rate $\mathcal{O}(t^{-(\beta-\alpha)/(\beta-\alpha+2)})$ and further derive Theorem 2. Besides, while classical theory assumes Assumption 8 holds for $\beta>1$ (i.e. $f_i\in(\mathcal{H}^\beta)^p\subset(\mathcal{H})^p$), our result relaxes this assumption to $\beta>\alpha$ and allows $\beta\leq 1$. Intuitively, this implies that the online stochastic approximation can address the misspecification case $f_i\notin(\mathcal{H})^p$ if $\alpha<1$.

[1] Allen-Zhu, Z., Li, Y., & Liang, Y. Learning and generalization in overparameterized neural networks, going beyond two layers. NIPS,2019.

[2] Ying, Y. and Pontil, M. Online gradient descent learning algorithms. Foundations of Computational Mathematics,2008.


[3] Lin, J. and Cevher, V. Optimal convergence for distributed learning with stochastic gradient methods and spectral algorithms. The Journal of Machine Learning Research,2020.

---

### Comment · Area_Chair_p5T9 · 2023-08-20
**Clarification questions**

Dear authors,

After going through your paper, the reviews, and your discussion with the reviewers, I would like your input on the following two points:

1. I could not find in your paper any examples of strongly monotone games with decision-dependent randomness motivated by real-world applications. You are mentioning ride-sharing markets in the introduction, but I was not able to find a precise description of how this model fits your strongly monotone framework - did I miss something in the paper or the appendix?

2. You state in L117 that the "loss functions $\ell_i$ are known to the agents" and only the "distribution maps $\mathcal{D}_i$ are unknown". However, the "bandit feedback" setting concerns the case where players only observe their realized losses and have no other information on the game - contrast for example with the paper of Narang et al (2022) where "*players have oracle access to queries of their loss function only, and therefore are faced with the problem of creating an estimate of their gradient from such queries*" (p. 19, beginning of Section 6.1). Could you explain how this fits the framework of Eq. (6), which requires access to the players' individual loss gradients $\nabla_i \ell_i$?

Thanks in advance for your input. Regards,

The AC

---

> ### Author Response · Authors · 2023-08-21
>
> Dear Area Chair,
>
> Thank you for taking the time to thoroughly review our paper. We appreciate your thoughtful consideration and are happy to provide clarifications on the points you have raised:
>
> **Regarding question 1:**
>
> We apologize for not including the real-world examples in the initial submission due to the limit of pages, we will present them in the final version of our paper. In the following, we present two examples to demonstrate decision-dependent games. We refer Narang et al. (2022) and [1] for more examples.
>
> **Example 1** (Revenue Maximization via Demand Forecasting). (Vignette 1, Narang (2022)) In the ride-sharing market, several platforms act as strategic agents (suppose there are $n$ platforms), predicting ride demands of strategic users in a city to maximize revenue. Typically, both drivers and passengers, regarded as strategic users, engage with multiple platforms by employing tactics such as "price shopping".  To elaborate, users call the ride in multiple platforms, and each platform $i$ presents its price and time cost (action $x_i$) for users. Strategic users compare prices and time costs among these platforms and choose the best one. Consequently, the forecasted ride demand $z_i$ for platform $i$, which is generated by the strategic users, relies on the platform's own decision $x_i$ as well as the choices of competitors $x_{-i}$, thereby shaping the distributions $z_i\sim\mathcal{D}_i(x)$.
>
> Suppose the revenue of the platform $i$ is given by the loss function $\ell_i(x,z_i) = -z_i^\top x_i+(\iota_i/2)\lVert x_i\rVert^2$, where $\iota_i\geq 0$ is some regularization parameter (Example 1 Narang (2022)). Under the parametric model (Assumption 1), $z_i$ takes the form $z_i = f_i(x)+\epsilon_i$, thus the performative gradient can be expressed by Eq. (5): $$\nabla_i\mathcal{L}_i(x) = \iota_i x_i-f_i(x)-\left(\frac{\partial f_i(x)}{\partial x_i}\right)^{\top}x_i.$$ Consequently, if parametric functions $f_i$ are Lipschitz continuous in $x$ and regularization parameters $\iota_i$ are sufficiently large, the proposed game is strongly monotone.
>
> **Example 2** (University Admissions). (Vignette 2, Narang (2022)) Multiple universities, acting as strategic agents,  evaluate applications to decide on admissions. Each applicant, considered a strategic user, tailors their application to meet the criteria of universities. Every university $i$ evaluates numerous applications, represented by data $z_i$ (might contain GPA and other related grades), and formulates a rule $x_i$ to decide which candidates are admitted. Each university's goal is to accept qualified students, and applicants may apply to various universities. To elaborate, students might compare different programs by assessing their admission rules and selecting several universities that match their qualifications. Consequently, the predicted applications $z_i$ received by the university $i$ are shaped by the joint rule $x$, thus formulating the decision-dependent distribution $z_i\sim\mathcal{D}_i(x)$. Furthermore, every university assesses the quality of students using a loss function, denoted as $\ell_i(x,z_i)$, and subsequently forms a decision-dependent game.
>
> **Regarding question 2:**
>
> In the context of decision-dependent games, the reward function $\mathcal{L}_i(x)$ for agent $i$ depends on the function $\ell_i(x,z_i)$ and the action-dependent distribution $\mathcal{D}_i(x)$ (which is determined by agents' joint action). To learn the Nash equilibrium, we need to know both $\ell_i$ and $\mathcal{D}_i(x)$. As detailed on page 4 (lines 136-145), learning  $\mathcal{D}_i(x)$ is the particular challenge in our game of decision-dependent distributions. In this regard, we only require a sample $z_i$ from $\mathcal{D}_i(x)$ when agents take the joint action $x$ (the estimation of the distribution $\mathcal{D}_i$ in iteration $t$ is carried out by solving the ERM version of Eq. (7), utilizing the samples $z_i^1,z_i^2,\cdots,z_i^{t}$ in conjunction with online stochastic approximation). **That is, we learn the action-dependent distribution from bandit feedback.** Besides, regarding the function $\ell_i$, we only require an oracle that outputs the noisy gradient
>
> $$\nabla_i\ell_i(x,z_i)+(\partial f_i^t(x)/\partial x_i)^\top\nabla_{z_i}\ell_i(x,z_i)$$ to conduct the projected gradient step (refer to Eq. (6) and line 171), which is common in the literature on first-order methods for learning games [2][3].
>
> [1] John Miller, Juan C Perdomo, and Tijana Zrnic. Outside the echo chamber: Optimizing the performative risk. arXiv preprint arXiv:2102.08570, 2021.
>
> [2] Panayotis Mertikopoulos, Mathias Staudigl. Equilibrium tracking and convergence in dynamic games. CDC 2021 - 60th IEEE Annual Conference on Decision and Control, 2021.
>
> [3] Duvocelle, B., Mertikopoulos, P., Staudigl, M., & Vermeulen, D. Multiagent online learning in time-varying games. Mathematics of Operations Research, 2023.

---

> > ### Comment · Area_Chair_p5T9 · 2023-08-21
> >
> > Thanks for the examples - in the meantime, I also found them in the paper of Narang et al.
> >
> > For the question on bandit feedback, your reply is missing my point: bandit feedback means that, if the players' profile is $x$, each player only gets to observe $\ell_i(x,z)$, with $z$ drawn according to $\mathcal{D}(x)$. As you mention in your reply, you require a (stochastic) gradient oracle for $\ell_i$ so, by definition, you are not working with bandit feedback, but with first-order feedback.
> >
> > Thanks again for your input. Regards,
> >
> > The AC

---

> > > ### Author Response · Authors · 2023-08-21
> > >
> > > Dear Area Chair,
> > >
> > > Thank you for pointing out the issue. We agree with your statement and apologize for the confusion. We will revise the final version of our paper to clarify that we assume the first-order oracle instead of bandit feedback, namely, that we assume the gradient oracle for the loss function $\ell_i$.
> > >
> > > In this regard, your comment motivates us to consider extending our method into the bandit feedback setting. Specifically, we might potentially extend [1] to decision-dependent games by approximating the gradient using noisy observations drawn from the values of the loss functions.
> > >
> > > Thanks again for your insightful comment.
> > >
> > > [1] Bravo, M., Leslie, D., & Mertikopoulos, P. (2018). Bandit learning in concave N-person games. Advances in Neural Information Processing Systems, 31.

---

### Decision · Program_Chairs · 2023-09-21

**Decision:**

Accept (poster)

**Comment:**

This paper examines the convergence of gradient descent in games with decision-dependent randomness (also referred to as multi-agent performative prediction). The authors consider a nonlinear model for the dependence of the randomness on the players' decision, and they analyze a kernel-based version of the online performative gradient descent (OPGD) algorithm, providing asymptotic rates of convergence for strongly monotone games.

This paper was discussed extensively by the reviewers, area chairs and senior area chairs. Initially, the reviewers were reluctant and raised a number of concerns, several of which were addressed by the authors' rebuttal, leading to a positive final assessment by the reviewers. At the same time, some concerns did remain, mostly regarding the paper's novelty over previous work by Narang et al.

The setting of Narang et al. is similar to the authors' except for the nonlinearity in the randomness, so the main issue was the degree of generality provided by the authors' model, and the new insights offered by the authors' analysis. Of the three novelties argued by the authors - different parametric dependence across agents, treating the infinite-dimensional RKHS setting, and the new convergence rates - the most significant ones concerned the intricacies of the RHKS treatment and the changes that this entails on the convergence rate of OPGD. In this regard, the program committee would urge the authors to focus their paper on introducing the Kernel version (section 4.2) and be clear that their main novelty is extending the prior linear parameterization case to a kernel-based one (which is what they analyze).

Another issue that came up in the discussion is that, contrary to the authors' claim in the abstract and the introduction, they do not treat the derivative-free / bandit feedback setting, as their algorithms requires access to a (possibly stochastic) first-order oracle. [By contrast, Narang et al. do treat the bona fide bandit feedback case] The discussion with the authors confirmed this point, so the authors would have to clarify that they are taking a first-order, oracle-based approach, not a zeroth-order, loss-based one.

Finally, even though the paper has been written in an overall clear manner, it lacks an in-depth discussion of applications relevant to ML. In this regard, the program committee would urge the authors to lay out a concrete application which can be based on examples from prior works, but which emphasizes the benefit of the kernel variant in an application context (e.g., maybe nonlinear pricing in a ride-share market...).

Despite the above limitations, the paper's analysis and results provide a number of fruitful insights into an active research field. In view of this, and conditioned on the above changes and revisions, I am happy to recommend acceptance.